# A GENERIC FRAMEWORK FOR CONFORMAL FAIRNESS

**Aditya T. Vadlamani** [1,*], **Anutam Srinivasan** [1,*], **Pranav Maneriker** [2,†],
**Ali Payani** [3], **Srinivasan Parthasarathy** [1]
[1]The Ohio State University   [2]Dolby Laboratories   [3]Cisco Systems

## ABSTRACT

Conformal Prediction (CP) is a popular method for uncertainty quantification with machine learning models. While conformal prediction provides probabilistic guarantees regarding the coverage of the true label, these guarantees are agnostic to the presence of sensitive attributes within the dataset. In this work, we formalize *Conformal Fairness*, a notion of fairness using conformal predictors, and provide a theoretically well-founded algorithm and associated framework to control for the gaps in coverage between different sensitive groups. Our framework leverages the exchangeability assumption (implicit to CP) rather than the typical IID assumption, allowing us to apply the notion of Conformal Fairness to data types and tasks that are not IID, such as graph data. Experiments were conducted on graph and tabular datasets to demonstrate that the algorithm can control fairness-related gaps in addition to coverage aligned with theoretical expectations.

## 1 INTRODUCTION

Machine learning (ML) models are increasingly used to make critical decisions in many fields of human endeavor, making it essential to quantify the uncertainty associated with their predictions. Conformal Prediction (CP) is a distribution-free framework (Vovk et al., 2005) which produces confidence sets with rigorous theoretical guarantees and has become popular in real-world applications (Cherian & Bronner, 2020). Post-hoc CP allows for facile integration into ML pipelines and applies to a wide variety of data types, including graph data (H. Zargarbashi et al., 2023; Huang et al., 2024), because of its weaker requirement of a *statistical exchangeability*.

Relatedly, ensuring the fairness of machine learning models is vital for their high-stakes deployments in critical decision-making. Biases affect ML models at different stages - from data collection to algorithmic learning stages (Mehrabi et al., 2021). During the data collection stage, measurement and representation biases can skew how each feature is interpreted, leading to inaccurate determinations by learning models. Algorithmic bias, caused by model design choices and prioritization of specific metrics while learning the model, can also lead to unfair outcomes. Many models inherit biases from historical outcomes (Kallus & Zhou, 2018; Dwork et al., 2012) and inadvertently skew decisions towards members of certain advantaged groups (Mehrabi et al., 2021). These biases have led to several global actors proposing and requiring practitioners to adhere to certain *fairness* standards (Hirsch et al., 2023). To facilitate ML pipeline and model adherence to socio-cultural or regulatory fairness standards, researchers have proposed methods to either construct fair-predictors (Alghamdi et al., 2022; Creager et al., 2019; Zhao et al., 2023) or audit fairness claims made by deployed machine learning models (Ghosh et al., 2021; Maneriker et al., 2023; Yan & Zhang, 2022).

However, these efforts on fairness (predictors, auditing, and uncertainty quantification) primarily focus on binary classification, often implicitly relying on the independent and identically distributed (IID) assumption, and largely do not bridge fairness and uncertainty quantification. The need to both quantify uncertainty and ensure fairness considerations are met is critical. A few researchers have started to examine how to assess (and possibly improve) the prediction quality of unreliable models (Wang & Wang, 2024) while meeting socio-cultural or regulatory standards of fairness. However, these efforts are limited in that they either require knowledge of group membership at inference time (a somewhat impractical assumption) (Lu et al., 2022) or are model-specific (Wang & Wang, 2024).

---

*Equal Contribution, †This work was done while the author was a student at The Ohio State University

**Key Contributions:** We propose a novel and comprehensive Conformal Fairness (CF) Framework to redress these concerns.

First, we develop the theoretical insights that facilitate how our framework leverages CP's distribution-free approach to build and construct fair uncertainty sets according to user-specified notions of fairness. Our framework is not only comprehensive but also highly flexible, as it can be adapted to bespoke user-specified fairness criteria. This adaptability ensures that the framework can be customized to meet the specific needs of different users, enhancing its practicality and usability.

Second, the weaker (exchangeability) assumptions required by CP allow us to extend the utility of our framework to fairness problems in graph models. Graph models, in particular, suffer from the *homophily effect*, which exacerbates inherent segregation due to node linkages and causes further biases in predictions (Current et al., 2022; Dong et al., 2023; He et al., 2023).

Third, we discuss how our approach serves as a fairness auditing tool for conformal predictors. This function is important as it allows one to verify model fairness, ensuring that fairness is not just a theoretical concept but a practical reality in predictive modeling.

Finally, we demonstrate the effectiveness of our CF Framework by evaluating fairness using multiple popular fairness metrics for multiple different conformal predictors on both real-world graph and tabular fairness datasets.

## 2 BACKGROUND

### 2.1 CONFORMAL PREDICTION

Conformal Prediction (Vovk et al., 2005) is a framework for quantifying the uncertainty of a model by constructing prediction sets that satisfy a *coverage* guarantee. For expository simplicity, we will focus on split (or inductive) conformal prediction (CP) in the classification setting. Given a calibration dataset, $\mathcal{D}_{\text{calib}} = \{(\boldsymbol{x}_i, y_i)\}_{i=1}^{n}$ and a test point $(\boldsymbol{x}_{n+1}, y_{n+1})$, where $\boldsymbol{x}_i \in \mathcal{X} = \mathbb{R}^d$ and $y_i \in \mathcal{Y} = \{0, \dots, K-1\}$, CP is used to construct a prediction set $\mathcal{C}(\boldsymbol{x}_{n+1})$ such that:

$$1 - \alpha \leq \text{Pr}[y_{n+1} \in \mathcal{C}(\boldsymbol{x}_{n+1})] \leq 1 - \alpha + \frac{1}{n+1}, \tag{1}$$

where $1 - \alpha \in (0, 1)$ is the coverage bound. Concretely, given a non-conformity score function $s : \mathcal{X} \times \mathcal{Y} \to \mathbb{R}$, let $\hat{q}(\alpha) = \text{Quantile}\left(\frac{\lceil (n+1)(1-\alpha) \rceil}{n}; \{s(\boldsymbol{x}_i, y_i)\}_{i=1}^{n}\right)$ be the *conformal quantile*. Then $\mathcal{C}_{\hat{q}(\alpha)}(\boldsymbol{x}_{n+1}) = \{y \in \mathcal{Y} : s(\boldsymbol{x}_{n+1}, y) \leq \hat{q}(\alpha)\}$ is a prediction set that satisfies Equation 1.

**Evaluating CP**: *Coverage* quantifies the true test time probability $\text{Pr}\left[y_{n+1} \in \mathcal{C}_{\hat{q}(\alpha)}(\boldsymbol{x}_{n+1})\right]$ while *efficiency* is the average test prediction set size, $\left|\mathcal{C}_{\hat{q}(\alpha)}(\boldsymbol{x}_{n+1})\right|$. Intuitively, there is an inverse relationship between coverage and efficiency, as a higher desired coverage is harder to achieve, the method may produce larger prediction sets to satisfy the guarantee. In CP, the only assumption made about the data is that $\mathcal{D}_{\text{calib}} \cup \{(\boldsymbol{x}_{n+1}, y_{n+1})\}$ is *exchangeable* – a weaker notion than iid, enabling its use on non-iid data, including graph data.

**Graph CP:** In this work, we focus on the node classification task. Given an attributed graph $\mathcal{G} = (\mathcal{V}, \mathcal{E}, \boldsymbol{X})$, where $\mathcal{V}$ is the set of nodes, $\mathcal{E}$ is the set of edges, and $\boldsymbol{X}$ is the set of node attributes. Let $\boldsymbol{A}$ be the adjacency matrix for the graph. Further, let $\mathcal{Y} = \{0, \dots, K-1\}$ denote the set of classes associated with the nodes. For $v \in \mathcal{V}$, $\boldsymbol{x}_v \in \mathbb{R}^d$ denotes its features and $y_v \in \mathcal{Y}$ denotes its true class. The task of node classification is to learn a model that predicts the label for each node given all the node features and the adjacency matrix, i.e., $(\boldsymbol{X}, \boldsymbol{A}, v) \mapsto y_v$. In the transductive setting, the entire graph, including test points, is accessible during the base model training. In this scenario, for any trained permutation-equivariant function (e.g., GNN) trained on a set of training/validation nodes, the scores produced on the calibration set and test set are exchangeable, thus enabling CP to be applied (H. Zargarbashi et al., 2023; Huang et al., 2024).

### 2.2 FAIRNESS METRICS

Group (or statistical) fairness requires individuals from different sensitive groups to be treated equally. Sensitive groups are subpopulations characterized by sensitive attributes, including gen-

der, race, and/or ethnicity. Group fairness metrics aim to observe bias in the predictions of a model between the different groups in a dataset. This work considers several popular fairness metrics, including equal opportunity, equalized odds, demographic parity, predictive equality, and predictive parity. For generality, we define the metrics for the multiclass setting with an $n$-ary sensitive attribute. Let $\mathcal{Y}^+$ denote the set of advantaged labels (e.g., "is_approved" in a loan approval task), $Y$ be the true label, and $\hat{Y}$ be the predicted label from a classifier. Let $\mathcal{G}$ be the set of all groups for the sensitive attribute(s). Table A1 discusses the formal definitions of different fairness metrics considered in this work.

Achieving *exact fairness* (i.e., the equality in Table A1) can be challenging or, in some cases, impossible (Barocas et al., 2023). Often, regulatory requirements focus on the difference (or ratio) in probabilities between groups for any given positive label. This is achieved by ensuring the difference (or ratio) meets a prespecified *closeness criterion*. For example, many regulatory bodies consider the **Four-Fifths Rule** (EEOC, 1979; Feldman et al., 2015), which asserts that the ratio of the selection probabilities between groups is at least $0.8$.

## 3 CONFORMAL FAIRNESS (CF) FRAMEWORK

In this section, we propose a theoretically well-founded framework using conformal predictors to control for fairness disparity between different sensitive groups. The framework is motivated by adapting the standard CP algorithm to determine conditional coverage *given* a score threshold, $\lambda$, for the prediction sets (i.e. $\mathcal{C}_\lambda(\boldsymbol{x}_{n+1}) = \{y \in \mathcal{Y} \mid s(\boldsymbol{x}_{n+1}, y) \leq \lambda\}$). Depending on the fairness metric, fairness disparity refers to gaps in group-conditional or group-and-class-conditional coverages between groups and advantaged labels. The conditional coverages are leveraged to evaluate if fairness is achieved for some closeness criterion $c$ for different fairness metrics. This is achieved by searching a *threshold space* $\Lambda$ for an optimal threshold $\lambda_{opt}$ that achieves the closeness criteria. The framework also handles user-defined metrics as discussed in Section 3.4, thus controlling for quantities, potentially orthogonal to conditional coverage.

### 3.1 EXEMPLAR CONFORMAL FAIRNESS (CF) METRICS

For conformal fairness, we adapt popular fairness metrics defined for *multiclass classification* (shown in Table A1). For standard point-wise predictions, fairness measures are concerned with the probability a prediction is a specific label (i.e., $\tilde{y} = \hat{Y}$), given a condition, i.e., $X \in g_a, Y = \tilde{y}$ for Equal Opportunity, for a particular covariate $(X, Y)$. We replace equivalence to the predicted value with set membership $(\tilde{y} \in \mathcal{C}_\lambda(X))$ to adapt these notions for prediction sets. The adapted conformal fairness metrics are in Table 1.

Table 1: Conformal Fairness Metrics.

| Metric | Definition |
|---|---|
| Demographic (or Statistical) Parity | $\left\| \Pr\left[\tilde{y} \in \mathcal{C}_\lambda(X) \,\middle\|\, X \in g_a\right] - \Pr\left[\tilde{y} \in \mathcal{C}_\lambda(X) \,\middle\|\, X \in g_b\right] \right\| < c, \; \forall g_a, g_b \in \mathcal{G}, \; \forall \tilde{y} \in \mathcal{Y}^+$ |
| Equal Opportunity | $\left\| \Pr\left[\tilde{y} \in \mathcal{C}_\lambda(X) \,\middle\|\, Y = \tilde{y}, X \in g_a\right] - \Pr\left[\tilde{y} \in \mathcal{C}_\lambda(X) \,\middle\|\, Y = \tilde{y}, X \in g_b\right] \right\| < c, \; \forall g_a, g_b \in \mathcal{G}, \; \forall \tilde{y} \in \mathcal{Y}^+$ |
| Predictive Equality | $\left\| \Pr\left[\tilde{y} \in \mathcal{C}_\lambda(X) \,\middle\|\, Y \neq \tilde{y}, X \in g_a\right] - \Pr\left[\tilde{y} \in \mathcal{C}_\lambda(X) \,\middle\|\, Y \neq \tilde{y}, X \in g_b\right] \right\| < c, \; \forall g_a, g_b \in \mathcal{G}, \; \forall \tilde{y} \in \mathcal{Y}^+$ |
| Equalized Odds | Equal Opp. and Pred. Equality |
| Predictive Parity | $\Pr\left[Y = \tilde{y} \,\middle\|\, \tilde{y} \in \mathcal{C}_\lambda(X), X \in g_a\right] = \Pr\left[Y = \tilde{y} \,\middle\|\, \tilde{y} \in \mathcal{C}_\lambda(X), X \in g_b\right], \; \forall g_a, g_b \in \mathcal{G}, \; \forall \tilde{y} \in \mathcal{Y}^+$ |

### 3.2 CONFORMAL FAIRNESS (CF) THEORY

Before presenting our framework, we first lay out the necessary theoretical groundwork. Detailed proofs are in Appendix B.

**Filtering $\mathcal{D}_{\textbf{calib}}$**: Group fairness metrics are evaluated on a subset of the population, defined by a condition on the data (i.e., membership in a group, true label value). For example, Demographic Parity is evaluated *per group* ($X \in g_a$ in definition), while Equal Opportunity is evaluated *per group **and** true label* ($Y = y, X \in g_a$ in definition). To formalize this notion, let $M$ denote

a fairness metric (e.g. Equal Opportunity) and define $F_M : \mathcal{X} \times \mathcal{Y} \times \mathcal{G} \times \mathcal{Y}^+ \to \{0, 1\}$ be a **filter function** which maps a calibration point along with a group and positive label, $(\boldsymbol{x}_i, y_i, g, \tilde{y})$, to 0 or 1 depending on whether the condition for the fairness metric, $M$, is satisfied. For Equal Opportunity, $F_M$ would instantiate to $F_{EO}(\boldsymbol{x}_i, y_i, g, \tilde{y}) := \mathbf{1}[\boldsymbol{x}_i \in g \cap y_i = \tilde{y}]$. We can filter $\mathcal{D}_{\text{calib}}$ to be $\mathcal{D}_{\text{calib}(g,\tilde{y})} = \{(\boldsymbol{x}_i, y_i) \in \mathcal{D}_{\text{calib}} \mid F_M(x_i, y_i, g, \tilde{y}) = 1\}$. By doing so, we provide guarantees regarding the conditional coverages as stated in Lemma 3.1

**Lemma 3.1.** *For any $(g, \tilde{y}) \in \mathcal{G} \times \mathcal{Y}^+$, calibrating on*
$\mathcal{D}_{\text{calib}(g,\tilde{y})} = \{(\boldsymbol{x}_i, y_i) \mid F_M(\boldsymbol{x}_i, y_i, g, \tilde{y}) = 1\}$ *guarantees the following about the conditional coverage:*

$$1 - \alpha \leq \Pr[y_{n+1} \in \mathcal{C}_\lambda(\boldsymbol{x}_{n+1}) \mid F_M(\boldsymbol{x}_{n+1}, y_{n+1}, g, \tilde{y}) = 1] \leq 1 - \alpha + \frac{1}{|\mathcal{D}_{\text{calib}(g,y)}| + 1} \quad (2)$$

*The **interval width** is $\frac{1}{|\mathcal{D}_{\text{calib}(g,y)}|+1}$.*

Prior work (Ding et al., 2024; Vovk et al., 2005; Lei et al., 2016) focused on the upper bound; however, the lower bound is also necessary for our framework.

**Inverse Quantile:** Given an $(1-\alpha)$-coverage level, we have that the the $(1-\alpha)$-quantile of the calibration non-conformity scores is the appropriate threshold to achieve Equation 1. For our framework, given a threshold, $\lambda$, we recover the coverage level. This can be done using the **inverse $\lambda$-quantile**. Formally, if $(\boldsymbol{x}_{n+1}, y_{n+1})$ is a test point and $\mathcal{S}_{\text{calib}} = \{s(\boldsymbol{x}_i, y_i) \mid (\boldsymbol{x}_i, y_i) \in \mathcal{D}_{\text{calib}}\}$, the inverse $\lambda$-quantile is given by

$$Q^{-1}(\lambda, \mathcal{S}_{\text{calib}}) := \Pr[s(\boldsymbol{x}_{n+1}, y_{n+1}) \leq \lambda] = \Pr[y_{n+1} \in \mathcal{C}_\lambda(\boldsymbol{x}_{n+1})].$$

Moreover, $Q^{-1}(\lambda, \mathcal{S}_{\text{calib}})$ is the coverage level for the label $y_{n+1}$. Lemma 3.2 asserts that the coverage level is within a bounded interval of length $\frac{1}{|\mathcal{D}_{\text{calib}}|+1}$.

**Lemma 3.2.** *For $\lambda \in [0, 1]$ and $n = |\mathcal{D}_{\text{calib}}|$,*

$$\frac{\sum_{i=1}^n \mathbf{1}[s(\boldsymbol{x}_i, y_i) \leq \lambda]}{n+1} \leq \Pr[y_{n+1} \in \mathcal{C}_\lambda(\boldsymbol{x}_{n+1})] \leq \frac{\sum_{i=1}^n \mathbf{1}[s(\boldsymbol{x}_i, y_i) \leq \lambda] + 1}{n+1}, \quad (3)$$

**CF for a Fixed Label:** In standard CP, coverage is only evaluated for the true label, $y_i$. However, for fairness evaluation, it is essential to balance disparity between groups for **all** positive labels (see Table A1. So for conformal fairness evaluation, coverage needs to be balanced between groups for any given $\tilde{y} \in \mathcal{Y}^+$, as seen in Table 1. Lemma 3.3 asserts that we can perform CP using a fixed label and get the same coverage guarantees.

**Lemma 3.3.** *Equation 1 holds if we replace $\{(\boldsymbol{x}_i, y_i)\}$ with $\{(\boldsymbol{x}_i, \tilde{y})\}$ for a fixed $\tilde{y} \in \mathcal{Y}$.*

**Connecting Theory to the Framework:** For a particular fairness metric, we filter the calibration set based on the conditional from Table 1 and achieve bounds on the conditional coverage with Lemma 3.1. By Lemma 3.3, the bounds continue to hold when considering the conditional coverage for a fixed positive label. We use Lemma 3.2 to perform an inverse quantile to compute the coverage under various $\lambda$ thresholds. With the coverages for a fixed positive label and each sensitive group, we compute the worst pairwise coverage gap across the groups using the bounds given by Lemma 3.3 to evaluate and control fairness at the desired closeness criterion.

### 3.3 CORE CONFORMAL FAIRNESS (CF) ALGORITHM

**Input**: The input to the core CF algorithm (Algorithm 1), include the calibration set, $\mathcal{D}_{\text{calib}}$, the set of labels ($\mathcal{Y}$) and positive labels ($\mathcal{Y}^+$), the set of sensitive groups, $\mathcal{G}$, a closeness criterion, $c$, the threshold search space, $\Lambda$, a fairness metric, $M$, and a corresponding filter function, $F_M$.

**Specifying $c$**: In practice, the choice of closeness criterion, $c$, may not be in the hands of the practitioner but instead defined by a (external) regulatory framework. For example, for Demographic Parity and $c$, we want that $\forall g_a, g_b \in \mathcal{G}$,

$$|\Pr[y_{n+1} \in \mathcal{C}_\lambda(\boldsymbol{x}_{n+1}) \mid \boldsymbol{x}_{n+1} \in g_a] - \Pr[y_{n+1} \in \mathcal{C}_\lambda(\boldsymbol{x}_{n+1}) \mid \boldsymbol{x}_{n+1} \in g_b]| < c.$$

**Choosing $\Lambda$**: The algorithm accepts a user-provided search space, $\Lambda$, which avoids degenerate thresholds and can guarantee desirable conditions. For our experiments, we set $\Lambda =$

$[\hat{q}(\alpha), \max\{\mathcal{S}_{\text{calib}}\}]$, ensuring that the optimal threshold, $\lambda_{opt}$, is at least $\hat{q}(\alpha)$. Since $\lambda_{opt} \geq \hat{q}(\alpha)$, the coverage increases for larger thresholds and still satisfies the $1 - \alpha$ coverage requirement. That is,

$$1 - \alpha \leq \Pr\big[y_{n+1} \in \mathcal{C}_{\hat{q}(\alpha)}(\boldsymbol{x}_{n+1})\big] \leq \Pr\big[y_{n+1} \in \mathcal{C}_{\lambda_{opt}}(\boldsymbol{x}_{n+1})\big].$$

**Procedure:** For each $\lambda \in \Lambda$, we want to check if it balances the coverage between groups for all positive labels. So, for each $(g, \tilde{y}) \in \mathcal{G} \times \mathcal{Y}^+$, we use $F_M$ to filter $\mathcal{D}_{\text{calib}}$ (Line 11 in Algorithm 1) and then compute the non-conformity scores, $\mathcal{S}_{\text{calib}_{(g,\tilde{y})}}$ (Line 12). With the inverse quantile, the coverage level is computed at the $\lambda$ threshold on the scores (Line 14). We then compare the coverages for a fixed $y \in \mathcal{Y}^+$ between groups and check if the worst-case disparity satisfies the desired closeness criterion (Lines 16-21), forming the set $\boldsymbol{\Lambda_M}$ (Line 2). We choose $\lambda_{opt} = \min_\lambda \Lambda_M$ (Line 3) to minimize the final prediction set size (i.e., get the best efficiency). When evaluating multiple fairness metrics simultaneously, for example with Equalized Odds, the framework can be used to construct the set of satisfying lambdas for Equal Opportunity and Predictive Equality, $\Lambda_{EO}$ and $\Lambda_{PE}$ respectively. Then, $\lambda_{opt} = \min_\lambda\{\Lambda_{EO} \cap \Lambda_{PE}\}$.

---

**Algorithm 1** Conformal Fairness Framework

---

1: **procedure** CONFORMAL_FAIRNESS($\mathcal{D}_{\text{calib}}$, $\mathcal{Y}$, $\mathcal{Y}^+$, $\mathcal{G}$, $c$, $\Lambda$, $F_M$)
2:     $\Lambda_M = \big\{\lambda \in \Lambda \mid \text{SATISFY\_LAMBDA}(\mathcal{D}_{\text{calib}}, \mathcal{Y}, \mathcal{Y}^+, \mathcal{G}, c, \lambda, F_M)\big\}$     ▷ Optimize for fairness
3:     $\lambda_{\text{opt}} = \min_\lambda \Lambda_M$     ▷ Optimize for efficiency
4:     **return** $\lambda_{\text{opt}}$
5: **end procedure**
6:
7: **procedure** SATISFY_LAMBDA($\mathcal{D}_{\text{calib}}$, $\mathcal{Y}$, $\mathcal{Y}^+$, $\mathcal{G}$, $c$, $\lambda$, $F_M$)
8:     label_coverages = $[0]_{(\mathcal{G}_i,y)\in\mathcal{G}\times\mathcal{Y}}$
9:     interval_widths = $[0]_{(\mathcal{G}_i,y)\in\mathcal{G}\times\mathcal{Y}}$
10:     **for** $(g, \tilde{y}) \in \mathcal{G} \times \mathcal{Y}^+$ **do**
11:         $\mathcal{D}_{\text{calib}_{(g,\tilde{y})}} = \{(\boldsymbol{x}_i, y_i) \in \mathcal{D}_{\text{calib}} \mid F_M(\boldsymbol{x}_i, y_i, g, \tilde{y}) = 1\}$
12:         $\mathcal{S}_{\text{calib}_{(g,\tilde{y})}} = \big\{s(\boldsymbol{x}_i, y_i) \mid (\boldsymbol{x}_i, y_i) \in \mathcal{D}_{\text{calib}_{(g,\tilde{y})}}\big\}$
13:         interval_widths$[(g, \tilde{y})] = \frac{1}{|\mathcal{D}_{\text{calib}_{(g,\tilde{y})}}|+1}$     ▷ Uses Lemma 3.1
14:         label_coverages$[(g, \tilde{y})] = Q^{-1}(\lambda, \mathcal{S}_{\text{calib}_{(g,\tilde{y})}})$     ▷ Uses Lemma 3.2
15:     **end for**
16:     **for** $\tilde{y} \in \mathcal{Y}^+$ **do**     ▷ Uses Lemma 3.3
17:         $\alpha_{\min} = \min(\text{label\_coverages}[(\cdot, \tilde{y})] - \text{interval\_widths}[(\cdot, \tilde{y})])$
18:         $\alpha_{\max} = \max(\text{label\_coverages}[(\cdot, y)])$
19:         **if** $\alpha_{\max} - \alpha_{\min} > c$ **then return** False
20:         **end if**
21:     **end for**
22:     **return** True
23: **end procedure**

---

**Using multiple $\lambda$ thresholds:** We also consider a classwise approach where we choose a $[\lambda_{opt}^0, \ldots, \lambda_{opt}^{k-1}] = \boldsymbol{\lambda_{opt}} \in [0,1]^K$ for each of the $K$ classes. $\lambda_{opt}^i$ is only required to satisfy the closeness criterion for the $i^{\text{th}}$ class. One can achieve this by setting $\mathcal{Y}^+ = \{\tilde{y}\}$ and repeating Lines 2 and 3 in Algorithm 1 for each $\tilde{y} \in \mathcal{Y}^+$. This allows for smaller $\lambda_{opt}^i$ to be chosen for most classes as they are no longer impacted by minority classes, which require a larger threshold to meet the closeness criterion.

A distinguishing feature of the CF framework is that it does not require group information at inference time. Though one can choose a different $\lambda$ for each $(g, y) \in \mathcal{G} \times \mathcal{Y}$ pair, in streaming (or online) settings, the sensitive attribute may be unavailable. For example, loan applications may be race or gender-blind to enforce fairer judgment. In these settings, the CF Framework is not limited and provides group conditional coverage when group information is absent at inference time.

### 3.4 FRAMEWORK EXTENSIBILITY

Algorithm 1 directly applies to Demographic Parity, Equal Opportunity, Predictive Equality, and Equalized Odds. The following modifications are necessary to accommodate Disparate Impact, Predictive Parity, and some user-defined metrics.

**Disparate Impact:** The standard criterion for Disparate Impact is the *Four-Fifths Rule* applied to Demographic Parity. To control the conditional coverages for the Four-Fifths Rule, we only change Line 19 in Algorithm 1 to check if $(1 - \alpha_{\max})/(1 - \alpha_{\min}) < c$ for $c = 0.8$.

**Predictive Parity:** Predictive Parity seeks to balance the Positive Predictive Value (PPV) between groups (Verma & Rubin, 2018). It differs from the other fairness metrics in Table 1 as it is *conditioned on membership in the prediction set*. Given the objective of balancing conditional coverage, the conformal definition of Predictive Parity, and Bayes' Theorem, we get

$$\Pr[Y = \tilde{y} \mid \tilde{y} \in \mathcal{C}_\lambda(X), X \in g_i] = \underbrace{\frac{\Pr[\tilde{y} \in \mathcal{C}_\lambda(X) \mid Y = \tilde{y}, X \in g_i]}{\Pr[\tilde{y} \in \mathcal{C}_\lambda(X) \mid X \in g_i]}}_{\text{Equal Opportunity over Demographic Parity}} \cdot \underbrace{\Pr[Y = \tilde{y} \mid X \in g_i]}_{\text{Conditional Label Probability}} \quad (4)$$

for $\tilde{y} \in \mathcal{Y}^+$ and $g_i \in \mathcal{G}$. A threshold, $\lambda$, is guaranteed to exist for any $\mathcal{Y}^+ \subseteq \mathcal{Y}$ if $c$ is greater than the maximum pairwise total variation distance of the group-conditioned label distribution. This is formalized in Theorem 3.4.

**Theorem 3.4.** *Let $W$ be a random variable for a label distribution over $\mathcal{Y}$. Let $W_i \sim W|(X \in g_i)$ – the label distribution conditioned on group membership. Then there exists $\lambda$ such that for $c \geq \max\{D_{TV}(W_i, W_j) \mid i, j \in \{1, \ldots, |\mathcal{G}|\}\}$, where $D_{TV}$ is the total variation distance[1], the difference in Predictive Parity between groups is within $c$.*

In Equation 4, the Equal Opportunity, Demographic Parity, and Conditional Label Probability terms lie within finite intervals. This allows us to compute an interval where conformal Predictive Parity holds and use the CF framework to identify values of $\lambda$ that meet the coverage closeness criterion. Further theoretical details and the proof of Theorem 3.4 are provided in Appendix C.

To control for arbitrarily small values of $c$, we use the *Predictive Parity Proxy*–an example of a user-defined metric– defined in Equation 5. For all $g_i \in \mathcal{G}, \tilde{y} \in \mathcal{Y}^+$,

$$\Pr(Y = \tilde{y} \mid \tilde{y} \in \mathcal{C}_\lambda(X), X \in g_i) - \Pr(Y = \tilde{y} \mid X \in g_i). \quad (5)$$

In cases where it is possible to assume the label distribution is independent of group membership, Equation 4 can be directly controlled for an arbitrarily small closeness criterion, $c$. Proofs and technical details on these modifications can be found in Appendix C.

### 3.5 Leveraging the CF Framework for Fairness Auditing

Using the Conformal Fairness Framework, one can audit if the disparity of a conformal predictor between multiple groups violates a user-specified fairness criterion. Specifically, we have thus far focused on fairness criteria concerning bounding the disparity between groups using the fairness metrics described in Table 1 by some closeness criterion, $c$. It is straightforward to support user-defined fairness metrics concerning label coverage. While Algorithm 1, as presented, gives a method of finding an optimal $\lambda$ threshold which satisfies the fairness guarantees using Lemmas 3.1, 3.2, and 3.3, the same SATISFY_LAMBDA procedure can be leveraged to check if a *given* $\lambda$ used by a conformal predictor satisfies the same fairness guarantees. Notably, the CF framework can also be leveraged even if the conformal predictor is treated as a black-box model. In this case, we construct an $\mathcal{D}_{\text{audit}}$ set exchangeable with the calibration data used for the conformal predictor. Using $\mathcal{D}_{\text{audit}}$, we can determine if the conformal predictor satisfies the corresponding fairness guarantee given the fairness metric and the $\lambda$ threshold used.

### 3.6 Non-Conformity Scores

There are several choices for the non-conformity score for performing fair conformal prediction with classification tasks. We currently implement TPS (Sadinle et al., 2019), APS (Romano et al., 2020), RAPS (Angelopoulos et al., 2022), DAPS (H. Zargarbashi et al., 2023), and CFGNN (Huang et al., 2024) in the CF framework, though any non-conformity score can be used. More details on the specifics of each non-conformity score can be found in Appendix D.2.

---

[1]A *modified total variation distance*, $D_{TV}^+(W_i, W_j) \coloneqq \sup_{k \in \mathcal{Y}^+} |\Pr[W_i = k] - \Pr[W_j = k]|$, can be used in place of $D_{TV}$ in Theorem 3.4 for a weaker assumption about $c$, which still gives a satisfying $\lambda$.

## 4 EXPERIMENTS

### 4.1 SETUP

**Datasets:** To evaluate the CF Framework, we used five multi-class datasets: Pokec-n (Takac & Zabovsky, 2012), Pokec-z (Takac & Zabovsky, 2012), Credit (Agarwal et al., 2021), ACSIncome (Ding et al., 2021), and ACSEducation (Ding et al., 2021) (see Table 2 for details). For each dataset, we use a $30\%/20\%/25\%/25\%$ stratified split of the labeled points for $\mathcal{D}_{\text{train}}/\mathcal{D}_{\text{valid}}/\mathcal{D}_{\text{calib}}/\mathcal{D}_{\text{test}}$.

Table 2: Dataset Statistics. T refers to Tabular, and G refers to Graph.

| Name | Type | Size | # Labeled | # Groups | # Classes |
|------|------|------|-----------|----------|-----------|
| ACSIncome | T | $1,664,500$ | ALL | race$(9)$ | 4 |
| ACSEducation | T | $1,664,500$ | ALL | race$(9)$ | 6 |

| Name | Type | $(|\mathcal{V}|, |\mathcal{E}|)$ | # Labeled | # Groups | # Classes |
|------|------|------|-----------|----------|-----------|
| Credit | T/G | $(30,000,\ 1,436,858)$ | ALL | age$(2)$ | 4 |
| Pokec-n | G | $(66,569,\ 729,129)$ | $8,797$ | region$(2)$, gender$(2)$ | 4 |
| Pokec-z | G | $(66,569,\ 729,129)$ | $8,797$ | region$(2)$, gender$(2)$ | 4 |

**Models:** For the graph datasets, we evaluated with GCN (Kipf & Welling, 2017), Graph-SAGE (Hamilton et al., 2017), or GAT (Veličković et al., 2018) as the base model (results reported are for the highest performing base model). For Credit, we additionally evaluated XGBoost (Chen & Guestrin, 2016) (i.e., ignoring the graph structure) as we empirically observed this approach to outperform the graph neural network baselines in terms of efficiency for this dataset. The choice of ignoring edge information while training Credit on XGBoost does not prohibit us from using CFGNN or DAPS, which utilize the edge information. The conformal predictor requires the softmax logits from the base model (i.e., XGBoost) but is otherwise model agnostic. For ACSIncome and ACSEducation, we used an XGBoost model. Each model's hyperparameters were tuned as discussed in Appendix D.3.

**Baseline:** For each dataset and CP non-conformity score, we built a conformal predictor to achieve a coverage level of $1 - \alpha = 0.9$. Then, we assess fairness according to the specific fairness metric using the SATISFY_LAMBDA from Algorithm 1 for $\lambda = \hat{q}(\alpha)$.

**Evaluation Metrics:** We report the *worst fairness disparity* and *efficiency*. For Disparate Impact, the worst fairness disparity is the *minimum* $(1 - \alpha_{\max})/(1 - \alpha_{\min})$ across the positive labels. For the remaining metrics, we record the *maximum* $\alpha_{\max} - \alpha_{\min}$ across the positive labels.

### 4.2 RESULTS

For each figure, we use a line to indicate the base conformal predictor's *average* worst fairness disparity across different thresholds, the bar plot for the worst fairness disparity using the CF Framework, and a dot to denote the desired fairness disparity. We report the average base performance for simplicity and readability of the figures. In every experiment, except for Figure 2, the CF framework was **better** than the average base conformal predictor. We provide a more granular version of Figure 2 with Figure E4, where it is clear that the framework performs better for every closeness threshold.

**Controlling for Fairness Disparity:** For different closeness thresholds, our CF Framework effectively controls the fairness disparity for several metrics compared to the base conformal predictor. In Figure 1 and 2, we can observe that in terms of fairness disparity, our CF Framework **precisely** (note step-wise change with $c$ on violations) improves upon the baseline conformal predictor. As with algorithmic fairness, a trade-off is involved in that there is a slightly worse efficiency. From Figure 2, we continue to observe this for both standard and graph-based conformal predictors. Furthermore, if the base conformal predictor is already "fair" according to our fairness disparity criterion, then the CF Framework will report the results accordingly. This phenomenon is observed with the CFGNN results in Figure 2, where the CF Framework matches the baseline regarding both evaluation metrics. This behavior of the CF Framework makes it suitable to leverage for black box fairness auditing (as noted previously). We present additional results, for example, the disparity results for the CF

Framework without classwise lambdas in Appendix E. Notably, the prediction set sizes are more prominent due to selecting a larger $\lambda$ than the classwise approach (see Figure E3 vs Figure 1).

**Controlling for Disparate Impact:** For Disparate Impact, we present results for the standard *Four-Fifths Rule*. In Table 3, we see that using the CF Framework can significantly improve the base conformal predictor for the *Four-Fifths Rule*. The disparate impact value is far below the desired $0.8$ for the base conformal predictor, sometimes even less than $0.4$, as with Credit with TPS and ACSIncome dataset. Our framework, however, is close to the $0.8$ value and in some cases surpasses it, like in Credit with CFGNN, with minor effects on the efficiency for both datasets.

Table 3: *Four-Fifths Rule* for Credit and ACSIncome. Our framework surpasses the base conformal predictor and achieves close to or exceeds the disparate impact value of $0.80$. The - means N/A.

| | | APS | | RAPS | | TPS | | CFGNN | | DAPS | |
| --- | --- | --- | --- | --- | --- | --- | --- | --- | --- | --- | --- |
| | | Base | CF | Base | CF | Base | CF | Base | CF | Base | CF |
| **Credit** | Disp. Impact | 0.646 | **0.821** | 0.586 | **0.768** | 0.252 | **0.793** | **0.922** | **0.922** | 0.539 | **0.809** |
| | Efficiency | 2.326 | 2.513 | 2.326 | 2.509 | 2.268 | 2.558 | 2.202 | 2.202 | 2.254 | 2.526 |
| **ACSIncome** | Disp. Impact | 0.397 | **0.797** | 0.387 | **0.790** | 0.356 | **0.798** | - | - | - | - |
| | Efficiency | 2.212 | 2.674 | 2.169 | 2.752 | 2.109 | 2.679 | - | - | - | - |

**Agnostic to Non-Conformity Score:** As discussed earlier, the CF Framework can support a variety of non-conformity scores, emphasizing the agnostic nature of our framework. We achieved effective results for conformal predictors with different underlying non-conformity score functions for all the experiments. Further results can be found in Appendix E.

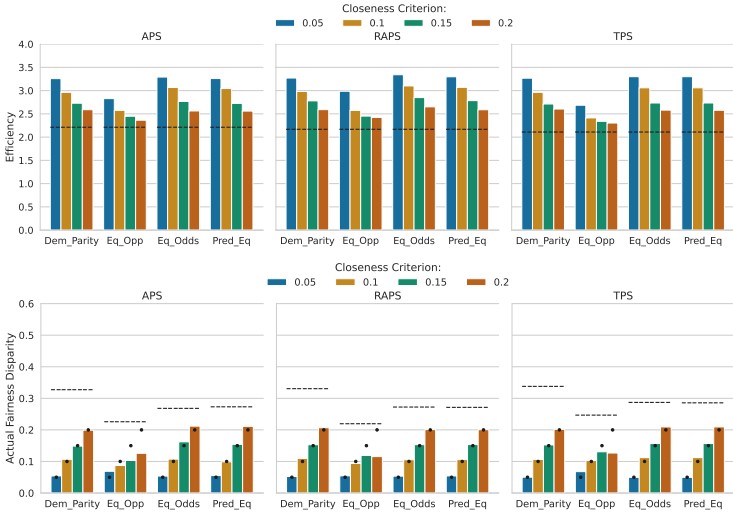

Figure 1: **ACSIncome**. The left two plots are efficiency results, while the right two are the fairness disparities for (a) APS, (b) RAPS, and (c) TPS. In all cases, our framework gives results at or better than the desired threshold and better than the baseline.

**Intersectional Fairness:** When characterizing data points into groups, we are not limited to a single sensitive attribute. In many applications, there can be multiple sensitive attributes (e.g., race and gender) that need to be considered. Our CF Framework is not limited to analyzing a single sensitive attribute. To demonstrate this, we experiment with the Pokec-n dataset. Pokec-n has two sensitive attributes, namely *region* and *gender*. We treat each combination of region and gender as a separate sensitive group and apply the CF framework to control for fairness disparities. Figure 3 shows that the CF framework improves upon the base conformal predictor regarding fairness disparity. This improvement is starker with the graph-based conformal predictors, CFGNN, and DAPS, as seen in Figure 3 plots (b) and (c).

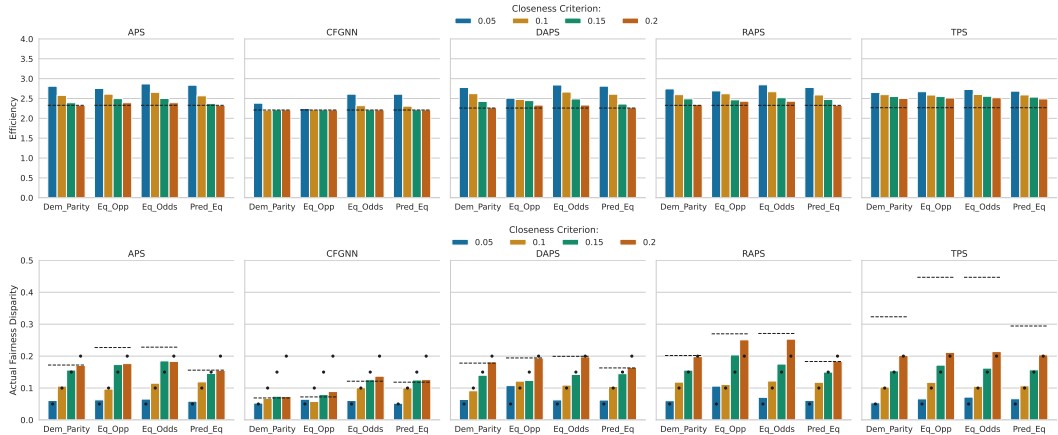

Figure 2: **Credit**. The top four plots are efficiency results, while the bottom four are the fairness disparities for (a) APS, (b) CFGNN, (c) DAPS, (d) RAPS, and (e) TPS. In all cases, our framework achieves the desired coverage gap better than the baseline, with a minor impact on efficiency.

A key challenge with intersectional fairness is the multiplicative increase in the number of groups (i.e., combinations of sensitive attributes and classes) that must be calibrated and evaluated. This increases the data requirements needed to satisfy the coverage guarantees discussed in Section 3.2, as these guarantees become harder to achieve when the size of $\mathcal{D}_{(g,y)}$ decreases. This problem is exacerbated (in empirical results) for datasets with only a few labeled points, such as Pokec-n. For Pokec-n, using a standard data split, the calibration set has around 2200 data points. The calibration set is then further split to get the conditional positive label coverage for each positive label and group pair. This results in the calibration being done with sets of fewer than a few hundred points, which is much lower than the suggested 1000 points in the literature (Angelopoulos & Bates, 2021). In Figure 3, the effect of this challenge is seen with the fairness disparity given by the CF Framework being slightly above the desired closeness threshold for $c = 0.1$. Nevertheless, the guarantees still hold, even under intersectional fairness constraints.

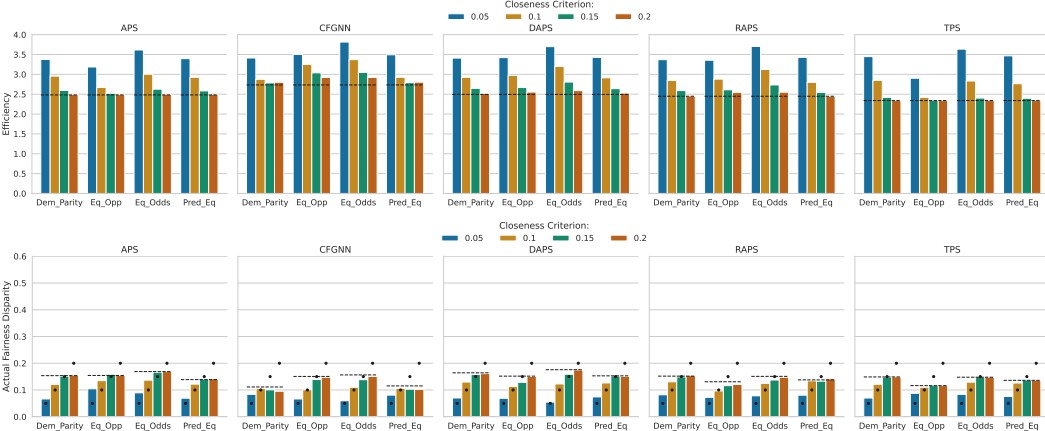

Figure 3: **Pokec-n** using **both** sensitive attributes. The top four plots are the efficiency results, while the bottom four are the fairness disparities for (a) APS, (b) CFGNN, (c) DAPS, (d) RAPS, and (e) TPS. CFGNN (b) and DAPS (c) achieve the desired fairness coverage thresholds better than standard CP methods.

**Predictive Parity Proxy:** As discussed, the CF framework is extensible to user-defined fairness notions. We consider the Predictive Parity Proxy in Equation 5 as an example of a user's ability to provide a reasonable fairness measure (Disparate Impact, above, is another example). An experiment on ACSEducation in Table 4 demonstrates we can control for arbitrarily small values of $c$, unlike the standard notion of Predictive Parity. Additionally, it empirically illustrates that we can control

for disparities of probabilities conditioned on the prediction set. This metric can also be applied in the graph setting, as seen in Appendix E.

Table 4: **ACSEducation**. The worst-case fairness disparity, based on the Predictive Parity Proxy, with our method is below the desired $c$ threshold, while the *average* baseline disparity is much higher ($> 0.30$) than all of the $c$ thresholds we consider.

|  | Closeness Threshold ($c$) | **0.05** | **0.10** | **0.15** | **0.20** | **Base (Average)** |
|---|---|---|---|---|---|---|
| **APS** | Max Fairness Disparity | 0.044 | 0.093 | 0.152 | 0.166 | 0.411 |
|  | Efficiency | 3.662 | 3.236 | 3.049 | 3.008 | 2.982 |
| **RAPS** | Max Fairness Disparity | 0.043 | 0.094 | 0.153 | 0.172 | 0.268 |
|  | Efficiency | 3.948 | 3.339 | 3.102 | 3.063 | 3.030 |
| **TPS** | Max Fairness Disparity | 0.038 | 0.091 | 0.167 | 0.199 | 0.319 |
|  | Efficiency | 3.662 | 3.061 | 2.880 | 2.845 | 2.828 |

### 4.3 DISCUSSION AND RELATED WORK

Few prior efforts study fairness and conformal prediction (Wang et al., 2024; Lu et al., 2022; Liu et al., 2022). One line of work has focused on applying fairness notions toward CP problems for regression tasks, explicitly focusing on Demographic Parity (Liu et al., 2022) and Equal Opportunity (Wang et al., 2024). Another line of work focuses on applying the notion of Overall Accuracy Equality for CP (Lu et al., 2022). This effort considers a specific medical application of detecting malignant skin conditions and applies group-balanced CP (Vovk, 2012).

An orthogonal direction is on (group) conditional CP. Foygel Barber et al. (2021) provides a theoretical grounding for conditional CP, while Gibbs et al. (2023) considers the impact of covariate shift for conditional coverage under the I.I.D assumption. Others Bastani et al. (2022); Jung et al. (2023) look at multivalid CP, which requires (1) group-conditional and (2) threshold-calibrated coverage guarantees – a distinct notion from Conformal Fairness. Deng et al. (2023) also introduces a generalization for multi-calibration and how it relates to algorithmic fairness for conformal prediction, focusing on equalized coverage for regression.

Our work differs in its breadth and flexibility (i.e., support for several fairness metrics and nonconformity scores) and focus on classification. Some existing works represent specific instantiations in our framework (e.g., Wang et al. (2024)). Others provide baselines for comparison (e.g., BatchGCP (Jung et al., 2023)), without our framework's theoretical guarantees for fairness. Appendix E.6 contains more details on BatchGCP and results. The CF framework generalizes group-balanced CP to consider the notion of coverage for specific labels, thus allowing us to evaluate disparity based on classical fairness metrics in a manner that does not require *a priori* knowledge of group membership at inference time, unlike many approaches listed above.

Lastly, CF can be used in fairness-critical domains where conditional conformal prediction is infeasible, such as finance, which can have strict fairness requirements (Agarwal et al., 2021), and health care (Wang et al., 2024), where privileged information may be unavailable at inference time.

## 5 CONCLUSION

In this work, we formalize the notion of Conformal Fairness (CF) for conformal predictors and propose a novel and comprehensive CF Framework. We provide a theoretically grounded algorithm that can be used to control for the gaps in conditional coverage, defined based on different fairness metrics, across sensitive groups. We conduct experiments on tabular and graph datasets, leveraging the exchangeability assumption of conformal prediction. We present results for CF based on various classical and user-defined fairness metrics on conformal predictors with various non-conformity score functions, including results on the framework's effectiveness in evaluating intersectional fairness with conformal predictors. We further describe how the CF framework can be practically leveraged for applications, including fairness auditing of conformal predictors. Future work could extend the framework to regression tasks and strengthen the theory by relaxing assumptions and exploring non-exchangeable settings.

ACKNOWLEDGMENTS

The authors acknowledge support from National Science Foundation (NSF) grant #2112471 (AI-EDGE) and a grant from Cisco Research (US202581249). Any opinions and findings are those of the author(s) and do not necessarily reflect the views of the granting agencies. The authors also thank the anonymous reviewers for their constructive feedback on this work.

REPRODUCIBILITY STATEMENT

In the spirit of reproducibility, the proofs for the theoretical aspects of our work can be found in Appendices B and C. Details of the experiments, including datasets, non-conformity scores, and hyperparameters, are provided in Appendix D. The source code is available at `https://github.com/AdityaVadlamani/conformal-fairness`.

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

## A   FAIRNESS METRICS

As discussed in Section 2.2, exact fairness is difficult to achieve. So, for many metrics, we want to say something about the difference between groups. Formally, for Demographic Parity, we may require that for some (small) $c \in (0, 1]$,

$$\max_{\tilde{y} \in \mathcal{Y}^+} \left\{ \left| \Pr\left[ \hat{Y} = \tilde{y} \mid X \in g_a \right] - \Pr\left[ \hat{Y} = \tilde{y} \mid X \in g_b \right] \right| \mid \forall g_a, g_b \in \mathcal{G} \right\} < c.$$

Similar requirements exist for the other fairness metrics.

Table A1: Fairness metrics formulations for multiclass classification (Rouzot et al., 2022).

| Metric | Definition |
|---|---|
| Demographic (or Statistical) Parity | $\Pr\left[ \hat{Y} = y \mid X \in g_a \right] = \Pr\left[ \hat{Y} = y \mid X \in g_b \right]$, $\forall g_a, g_b \in \mathcal{G}$, $\forall y \in \mathcal{Y}^+$ |
| Equal Opportunity | $\Pr\left[ \hat{Y} = y \mid Y = y, X \in g_a \right] = \Pr\left[ \hat{Y} = y \mid Y = y, X \in g_b \right]$, $\forall g_a, g_b \in \mathcal{G}$, $\forall y \in \mathcal{Y}^+$ |
| Predictive Equality | $\Pr\left[ \hat{Y} = y \mid Y \neq y, X \in g_a \right] = \Pr\left[ \hat{Y} = y \mid Y \neq y, X \in g_b \right]$, $\forall g_a, g_b \in \mathcal{G}$, $\forall y \in \mathcal{Y}^+$ |
| Equalized Odds | Equal Opp. and Pred. Equality |
| Predictive Parity | $\Pr\left[ Y = y \mid \hat{Y} = y, X \in g_a \right] = \Pr\left[ Y = y \mid \hat{Y} = y, X \in g_b \right]$, $\forall g_a, g_b \in \mathcal{G}$, $\forall y \in \mathcal{Y}^+$ |

## B   PROOFS

### B.1   PROOF OF LEMMA 3.1

**Lemma 3.1.** *For any $(g, \tilde{y}) \in \mathcal{G} \times \mathcal{Y}^+$, calibrating on*
$\mathcal{D}_{\mathrm{calib}(g,\tilde{y})} = \{ (\boldsymbol{x}_i, y_i) \mid F_M(\boldsymbol{x}_i, y_i, g, \tilde{y}) = 1 \}$ *guarantees the following about the conditional coverage:*

$$1 - \alpha \leq \Pr[y_{n+1} \in \mathcal{C}_\lambda(\boldsymbol{x}_{n+1}) \mid F_M(\boldsymbol{x}_{n+1}, y_{n+1}, g, \tilde{y}) = 1] \leq 1 - \alpha + \frac{1}{|\mathcal{D}_{\mathrm{calib}(g,y)}| + 1} \quad (2)$$

*The **interval width** is $\frac{1}{|\mathcal{D}_{\mathrm{calib}(g,y)}| + 1}$.*

*Proof.* The proof for the upper bound follows Ding et al. (2024), where the test score $s(\boldsymbol{x}_{n+1}, y_{n+1})$ follows the distribution of scores in $\mathcal{D}_{\mathrm{calib}} \cap \mathcal{R}$, thus holding the conditional coverage. The upper-bound guarantee directly follows Romano et al. (2019), assuming distinct non-conformity scores or a suitably random way to tie-break equal scores. $\square$

### B.2   PROOF OF LEMMA 3.2

To prove Lemma 3.2, we use the following Lemma that is stated in the proof of Theorem D.1 from Angelopoulos & Bates (2021):

**Lemma B.1.** *Suppose you have $n+1$ exchangeable random variables $Z_1, Z_2, \ldots, Z_n, Z_{n+1}$. Then, $\Pr[Z_{n+1} \leq Z_{(k)}] = \frac{k}{n+1}$ for all $k \leq n$, where $Z_{n+1}$ is the test point.*

*Proof Sketch.* Using $Z_1, \ldots, Z_n$, you can form $n + 1$ intervals, $(L, Z_{(1)}], (Z_{(1)}, Z_{(2)}], \ldots, (Z_{(n-1)}, Z_{(n)}], (Z_{(n)}, U)$, where $L$ and $U$ are the lower and upper bounds of the random variables. Exchangeability gives us that $Z_{n+1}$ falls in any of the $n + 1$ intervals with equal probability. Thus, $\Pr[Z_{n+1} \leq Z_{(k)}]$ is the probability that it falls in the first $k$ intervals giving us $\Pr[Z_{n+1} \leq Z_{(k)}] = \frac{k}{n+1}$. $\square$

Now using Lemma B.1, we will prove Lemma 3.2.

**Lemma 3.2.** *For $\lambda \in [0, 1]$ and $n = |\mathcal{D}_{\mathrm{calib}}|$,*

$$\frac{\sum_{i=1}^n \mathbf{1}[s(\boldsymbol{x}_i, y_i) \leq \lambda]}{n + 1} \leq \Pr[y_{n+1} \in \mathcal{C}_\lambda(\boldsymbol{x}_{n+1})] \leq \frac{\sum_{i=1}^n \mathbf{1}[s(\boldsymbol{x}_i, y_i) \leq \lambda] + 1}{n + 1}, \quad (3)$$

*Proof.* Let $s_i = s(\boldsymbol{x}_i, y_i)$ for brevity. By Lemma B.1, we have that $\Pr(s_{n+1} \leq s_{(k)}) = \frac{k}{n+1}$ for all $k \leq n$. Let $k$ be s.t. $s_{(k)} \leq \lambda \leq s_{(k+1)}$. We then have that $\Pr(s_{n+1} \leq s_{(k)}) \leq \Pr(s_{n+1} \leq \lambda) \leq \Pr(s_{n+1} \leq s_{(k+1)})$. So,

$$\frac{k}{n+1} \leq \Pr(s_{n+1} \leq \lambda) \leq \frac{k+1}{n+1}.$$

Since $s_1, \ldots, s_n$ are known empirically, we have that $k = \sum_{i=1}^n \mathbf{1}[s_i \leq \lambda]$. Hence,

$$\frac{\sum_{i=1}^n \mathbf{1}[s_i \leq \lambda]}{n+1} \leq \Pr(s_{n+1} \leq \lambda) \leq \frac{\sum_{i=1}^n \mathbf{1}[s_i \leq \lambda] + 1}{n+1}.$$

Since $\Pr[s_{n+1} \leq \lambda] = \Pr[y_{n+1} \in \mathcal{C}_\lambda(\boldsymbol{x}_{n+1})]$, we get Equation 3.

$\square$

### B.3 PROOF OF LEMMA 3.3

**Lemma 3.3.** *Equation 1 holds if we replace $\{(\boldsymbol{x}_i, y_i)\}$ with $\{(\boldsymbol{x}_i, \tilde{y})\}$ for a fixed $\tilde{y} \in \mathcal{Y}$.*

*Proof.* By computing $\hat{q}(\alpha) = \text{Quantile}\left(\frac{\lceil(n+1)(1-\alpha)\rceil}{n}; \{s(\boldsymbol{x}_i, \tilde{y})\}_{i=1}^n\right)$, where $\tilde{y}$ is the fixed label. Using the assumption of exchangeability we have that $s(\boldsymbol{x}_1, \tilde{y}), s(\boldsymbol{x}_2, \tilde{y}), \ldots, s(\boldsymbol{x}_{n+1}, \tilde{y})$ is an exchangeable sequence. Thus

$$1 - \alpha \leq \Pr[s(\boldsymbol{x}_{n+1}, \tilde{y}) \leq \hat{q}(\alpha)] \leq 1 - \alpha + \frac{1}{n+1}. \tag{6}$$

Equivalently,

$$1 - \alpha \leq \Pr\left[\tilde{y} \in \mathcal{C}_{\hat{q}(\alpha)}(x_{n+1})\right] \leq 1 - \alpha + \frac{1}{n+1}. \tag{7}$$

$\square$

## C FURTHER DISCUSSION ON PREDICTIVE PARITY

**Theorem 3.4.** *Let $W$ be a random variable for a label distribution over $\mathcal{Y}$. Let $W_i \sim W|(X \in g_i)$ – the label distribution conditioned on group membership. Then there exists $\lambda$ such that for $c \geq \max\{D_{TV}(W_i, W_j) \mid i, j \in \{1, \ldots, |\mathcal{G}|\}\}$, where $D_{TV}$ is the total variation distance[2], the difference in Predictive Parity between groups is within $c$.*

*Proof.* Let $\lambda$ be the maximum value it can take on as the threshold of the prediction set. Let $y \in \mathcal{Y}^+$ and $g_m, g_n \in \mathcal{G}$. Then,

$$\Pr\left[Y = y \mid y \in \mathcal{C}_\lambda(X), X \in g_m\right] = \frac{\Pr\left[y \in \mathcal{C}_\lambda(X) \mid Y = y, X \in g_m\right]}{\Pr\left[y \in \mathcal{C}_\lambda(X) \mid X \in g_m\right]} \Pr\left[Y = y \mid X \in g_m\right]$$

$$= \Pr\left[Y = y \mid X \in g_m\right]$$

since the numerator and the denominator are 1 due to the selection of $\lambda$. Using the same argument for $g_n$ and the definition of $D_{TV}$ the difference in Predictive Parities is:

$$\left|\Pr\left[Y = y \mid X \in g_m\right] - \Pr\left[Y = y \mid X \in g_n\right]\right| \leq D_{TV}(W_m, W_n)$$

Since, $D_{TV}(W_m, W_n) \leq \max\{D_{TV}(W_i, W_j) \mid i, j \in \{1, \ldots, |\mathcal{G}|\}\} \leq c$, the choosen $\lambda$ value causes the Predictive Parity difference to be less than $c$. Thus a solution exists for $c \geq \max\{D_{TV}(W_i, W_j) \mid i, j \in \{1, \ldots, |\mathcal{G}|\}\}$. $\square$

---

[2] A *modified total variation distance*, $D_{TV}^+(W_i, W_j) := \sup_{k \in \mathcal{Y}^+} |\Pr[W_i = k] - \Pr[W_j = k]|$, can be used in place of $D_{TV}$ in Theorem 3.4 for a weaker assumption about $c$, which still gives a satisfying $\lambda$.

## C.1 Achieving Arbitrary Closeness

We will further discuss the two methods for controlling for arbitrarily small values of $c$, depending on whether the label distribution is independent of group membership.

### C.1.1 Label Distribution Independent of Group Membership

Assuming independence, we have the following corollary of Theorem 3.4:

**Corollary C.1.** *Given a random variable, $W$, from a label distribution over $\mathcal{Y}$ that is independent of group membership, then the Conformal Fairness Framework can find a $\lambda$ such that the disparity of Predictive Parity is within any $c > 0$.*

*Proof.* Let $W_i \sim W | (X \in g_i)$ be the label distribution condition on group membership. Using the independence assumption we get $W = W_i$, $\forall i \in \{1, \ldots, |\mathcal{G}|\}$, thus $D_{TV}(W_i, W_j) = 0$, $\forall i, j \in \{1, \ldots, |\mathcal{G}|\}$. Hence, using Theorem 3.4, $c > \max\{D_{TV}(W_i, W_j) \mid i, j \in \{1, \ldots, |\mathcal{G}|\}\} = 0$. Thus a $\lambda$ can be found for any $c > 0$. $\qquad\square$

### C.1.2 Without Independence Assumption

When independence cannot be assumed, then we propose a proxy for Predictive Parity that the Conformal Fairness Framework can use. We propose a *Predictive Parity Proxy* as the balancing the quantity $\Pr(Y = y \mid y \in \mathcal{C}_\lambda(X), X \in g_a) - \Pr(Y = y \mid X \in g_a)$ across all groups and positive labels. Thus, for the framework, given a user-specified value for $c$, we want $\forall g_a, g_b \in \mathcal{G}$, $\forall y \in \mathcal{Y}^+$:

$$\left| (\Pr[Y = y \mid y \in \mathcal{C}_\lambda(X), X \in g_a] - \Pr[Y = y \mid X \in g_a]) \right. \\ \left. - (\Pr[Y = y \mid y \in \mathcal{C}_\lambda(X), X \in g_b] - \Pr[Y = y \mid X \in g_b]) \right| < c. \quad (8)$$

Observe that using $\lambda = \sup \Lambda$ will make Equation 8 equal to zero, thus $c$ can be arbitrarily small. Intuitively, this proxy balances the information provided about an outcome if the label is in the prediction set, similar to Predictive Parity. Formally, if the label distribution is independent of group membership, then balancing the proxy would be the same as balancing Predictive Parity.

## D Additional Experiment Details

### D.1 Datasets

**Credit (*G*):** The Credit dataset is from the `UCI` repository, and traditionally the binary target is to predict the existence of default payments (Yeh & Lien, 2009). We used a graph version of Credit as considered by Agarwal et al. (2021). To convert the dataset to a multi-class dataset, we used the education level (4 labels) as the target and used gender as the sensitive attribute, as done by Liu et al. (2023).

**Pokec-{n,z} (*G*):** The Pokec dataset (Takac & Zabovsky, 2012) is a social-network graph dataset collected from Pokec, a popular social network in Slovakia. Since several rows in the dataset are missing features, two commonly used subgraphs are the Pokec-z and Pokec-n datasets. The graphs have 4 labels, corresponding to the fieldwork. They also have two sensitive attributes, gender (2 groups) and region (2 groups). Our experiments consider each attribute individually and intersectional fairness by creating an attribute with 4 (2x2) groups.

**ACSIncome (*T*):** In the fairness space, the American Community Services (ACS) datasets from the `Folktables` library are widely used (Ding et al., 2021). For ACSIncome, we used the standard ACSIncome dataset in Foltables however, we divided the targets into 4 classes by evenly dividing the income into 4 brackets. Race is the sensitive attribute and has 9 groups.

**ACSEducation (*T*):** Similar to ACSIncome, we used the ACS data and selected the Education Level as our target. We broke the education level into 6 groups {did not complete high school,

has a high school diploma, has a GED, started an undergrad program, completed an undergrad program, and completed graduate or professional school}. ACSEducation also uses race as a sensitive attribute.

### D.2 NON-CONFORMITY SCORES

Let $\hat{\pi}$ be a trained classification model with softmaxed output.

**Threshold Prediction Sets (TPS)**   In TPS (Sadinle et al., 2019), the score function is $s(\boldsymbol{x}, y) = 1 - \hat{\pi}(\boldsymbol{x})_y$, where $\hat{\pi}(\boldsymbol{x})_y$ is the class probability for the correct class. This is the simplest method, which is also shown to be optimal with respect to efficiency (Sadinle et al., 2019).

**Adaptive Prediction Sets (APS)**   The most popular baseline when comparing CP method is APS (Romano et al., 2019). The scoring function works by sorting the softmax logits in descending order and accumulating the class probabilities until the correct class is included. For tighter prediction sets, randomization is introduced through a uniform random variable. Formally, if $\hat{\pi}(\boldsymbol{x})_{(1)} \geq \hat{\pi}(\boldsymbol{x})_{(2)} \geq \cdots \geq \hat{\pi}(\boldsymbol{x})_{(K-1)}$, $u \sim U(0,1)$, and $r_y$ is the rank of the correct label, then

$$s(\boldsymbol{x}, y) = \left[\sum_{i=1}^{r_y} \hat{\pi}(\boldsymbol{x})_{(i)}\right] - u\hat{\pi}(\boldsymbol{x})_y.$$

**Regularized Adaptive Prediction Sets (RAPS)**   One drawback of APS is that it can produce large prediction sets. Angelopoulos et al. (2022) introduces a regularization approach for APS. Given the same setup and notation as APS, define $o(\boldsymbol{x}, y) = |\{c \in \mathcal{Y} : \hat{\pi}(\boldsymbol{x})_y \geq \hat{\pi}(\boldsymbol{x})_c\}|$. Then,

$$s(\boldsymbol{x}, y) = \left[\sum_{i=1}^{r_y} \hat{\pi}(\boldsymbol{x})_{(i)}\right] - u\hat{\pi}(\boldsymbol{x})_y + \nu \cdot \max\{(o(\boldsymbol{x}, y) - k_{reg}), 0\},$$

where $\nu$ and $k_{reg} \geq 0$ are regularization hyperparameters.

**Diffusion Adaptive Prediction Sets (DAPS)**   Graphs are rich with neighborhood information, with nodes tending to be homophilous. Intuitively, this suggests that the non-conformity scores of connected nodes are also related. To utilize this observation, DAPS H. Zargarbashi et al. (2023) performs a one-step diffusion update on the non-conformity scores. Formally, if $s(\boldsymbol{x}, y)$ is a point-wise score function (e.g., APS), then the diffusion step gives a new score function

$$\hat{s}(\boldsymbol{x}, y) = (1 - \delta)s(\boldsymbol{x}, y) + \frac{\delta}{|\mathcal{N}_{\boldsymbol{x}}|} \sum_{\boldsymbol{u} \in \mathcal{N}_{\boldsymbol{x}}} s(\boldsymbol{u}, y),$$

where $\delta \in [0, 1]$ is a diffusion hyperparamter and $\mathcal{N}_{\boldsymbol{x}}$ is the 1-hop neighborhood of $\boldsymbol{x}$.

**Conformalized GNN (CFGNN)**   CFGNN (Huang et al., 2024) is a GNN approach to graph CP. The underlying observation is that the inefficiencies are correlated between nodes with similar neighborhood topologies. Bearing this in mind, using the calibration set, a second GNN is trained to correct the scores from the base model to optimize for efficiency through an inefficiency loss function Huang et al. (2024) propose. The inefficiency loss function definition includes a point-wise score function and can be different for training and validation. For our experiments, we set the score function to be APS and kept it consistent between training and validation.

### D.3 HYPERPARAMETER TUNING

Hyperparameter tuning was done using Ray Tune (Liaw et al., 2018). For the Pokec_n and Pokec_z datasets, hyperparameters for the base GNN models were tuned via random search using Table D1 for each model type (i.e., GCN, GAT, and GraphSAGE) and for each choice of the sensitive attribute(s). For the Credit, ACS Income, and ACS Education datasets, the base XGBoost models were tuned via random search using Table D2 for each choice of sensitive attribute(s).

For Credit, Pokec_n, and Pokec_z, we tune the hyperparameters for the CFGNN model via random search using Table D3 for each model type (e.g., GCN, GAT, and GraphSAGE), for each dataset, and choice of sensitive attribute(s). We set $\mathcal{D}_{\text{calib}} = \mathcal{D}_{\text{test}} = (1 - \mathcal{D}_{\text{train}} - \mathcal{D}_{\text{valid}})/2$.

All experiments were run on a single P100 GPU.

In the interest of reproducibility, the source code for the CF Framework is provided in the supplementary material.

Table D1: Hyperparameter search space for the base GNN model for Pokec_n and Pokec_z. The last two rows are layer-type specific for GAT and GraphSAGE, respectively.

| Hyperparameter | Search Space |
|---|---|
| batch_size | 64 |
| lr | $\text{loguniform}(10^{-4}, 10^{-1})$ |
| hidden_channels | $\{16, 32, 64, 128\}$ |
| layers | $\{1, 2, 4\}$ |
| dropout | $\text{uniform}(0.1, 0.8)$ |
| heads | $\{2, 4, 8\}$ |
| aggr_fn | $\{\text{mean, gcn, pool, lstm}\}$ |

Table D2: Hyperparameter search space for the base XGBoost model for Credit, ACS Education, and ACS Income.

| Hyperparameter | Search Space |
|---|---|
| lr | $\text{loguniform}(10^{-4}, 10^{-1})$ |
| n_estimators | $\{2, \ldots, 500\}$ |
| max_depth | $\{2, \ldots, 30\}$ |
| gamma | $\text{uniform}(0, 1)$ |
| colsample_bytree | $\text{uniform}(0.25, 1.0)$ |
| colsample_bylevel | $\text{uniform}(0.25, 1.0)$ |
| colsample_bynode | $\text{uniform}(0.25, 1.0)$ |
| subsample | $\text{uniform}(0.5, 1.0)$ |

Table D3: Hyperparameter search space for the CFGNN model for Credit, Pokec_n, and Pokec_z. The last two rows are layer-type specific for GAT and GraphSAGE, respectively.

| Hyperparameter | Search Space |
|---|---|
| batch_size | 64 |
| lr | $\text{loguniform}(10^{-4}, 10^{-1})$ |
| hidden_channels | $\{16, 32, 64, 128\}$ |
| layers | $\{1, 2, 3, 4\}$ |
| dropout | $\text{uniform}(0.1, 0.8)$ |
| $\tau$ | $\text{loguniform}(10^{-3}, 10^{1})$ |
| heads | $\{2, 4\}$ |
| aggr_fn | $\{\text{mean, gcn, pool, lstm}\}$ |

## E ADDITIONAL RESULTS

Here we provide additional results and discourse for each dataset and experiment we discuss in the main paper.

## E.1 ACSEDUCATION

Below we include results for the ACSEducation dataset, which is unique as it has a greater number of classes and is a custom dataset generated from the ACS datasets.

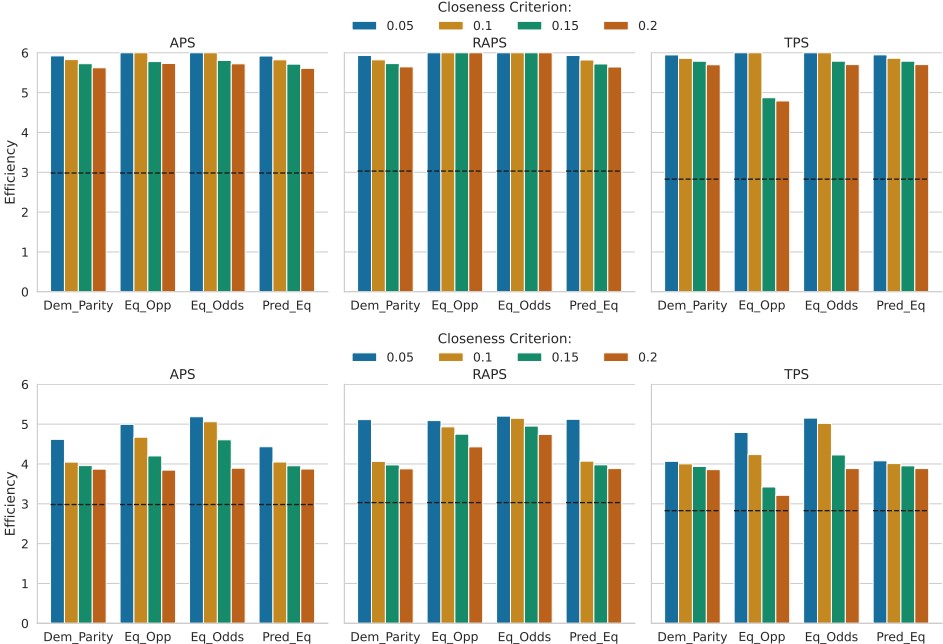

Figure E1: **ACSEducation.** Comparison of efficiencies when using the CF Framework without (top) and with (bottom) classwise lambdas. We observe that the efficiencies are better in the *right* plot. This is because $\forall_{i,...,k} \lambda_{\text{non-classwise}} \geq \lambda^i_{\text{classwise}}$ (k is the number of classes), which causes fewer labels to be included in the prediction set, thus improving efficiency with the classwise approach. For some experiments, the fairness disparity is 0 (e.g. APS and RAPS in the no-classwise setting), because the framework is producing the full prediction set–the trivial case–which means the coverage of $\tilde{y} \in \mathcal{Y}^+$ is 1.00, thus causing the disparity to be 0.

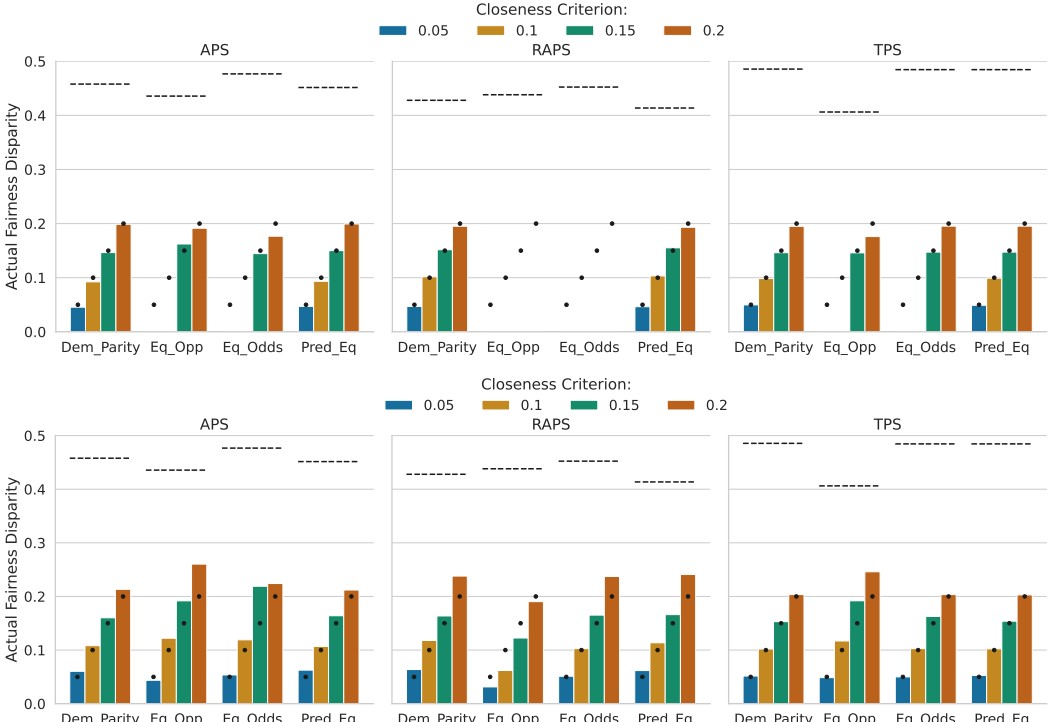

Figure E2: **ACSEducation.** Comparison of fairness disparities when using the CF Framework without (top) and with (bottom) classwise lambdas. We observe that the fairness disparities are better in the *left* plot. This is because by using a single $\lambda$, only the hardest-to-satisfy label will be at or around the coverage gap, $c$, unlike classwise which ensures all labels will be at or around the coverage gap, $c$. Since fewer labels have coverages around the coverage gap, for non-classwise in (left) the likelihood of being above the threshold is limited - as opposed to the classwise approach (right).

## E.2 Impact of Classwise Lambdas

### E.2.1 ACSIncome

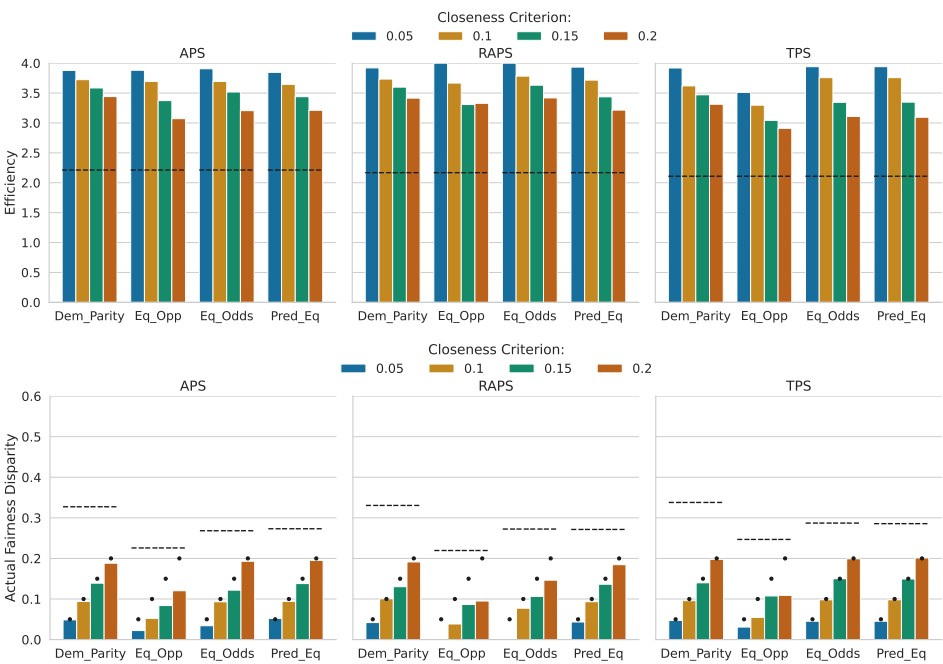

Figure E3: **ACSIncome.** The efficiency (top) and fairness disparity (bottom) plots when **not** using classwise lambdas. We observe that the efficiency is worse, but the disparity control is much better than using classwise lambdas (see Figure 1).

### E.2.2 Credit

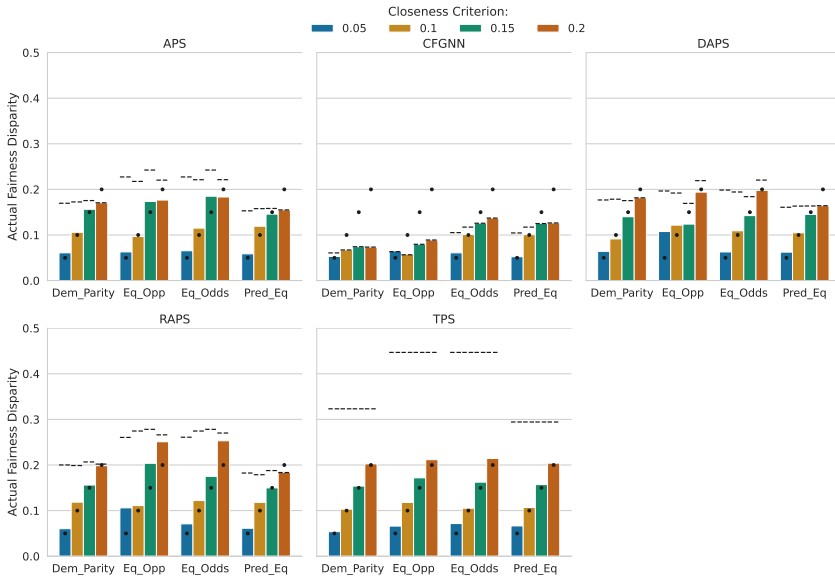

Figure E4: **Credit**. A granular version of Figure 2, which has a bar over each value of $c$ rather than considering an average. This plot clarifies that the CF framework matches or exceeds the baseline conformal predictor's performance in fairness disparity.

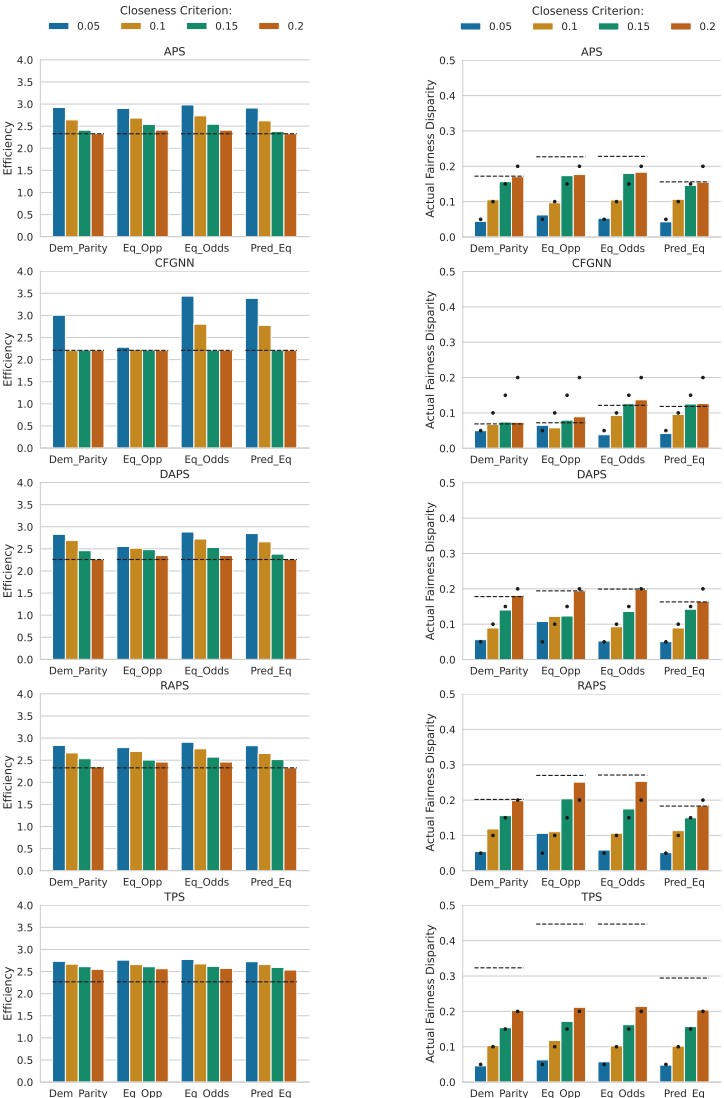

Figure E5: **Credit.** The efficiency (left) and fairness disparity (right) plots when **not** using classwise lambdas. We observe that the efficiency is worse, but the disparity control is much better than using classwise lambdas (see Figure 2).

### E.3 IMPACT OF DIFFERENT POKEC SENSITIVE ATTRIBUTES

#### E.3.1 POKEC-N

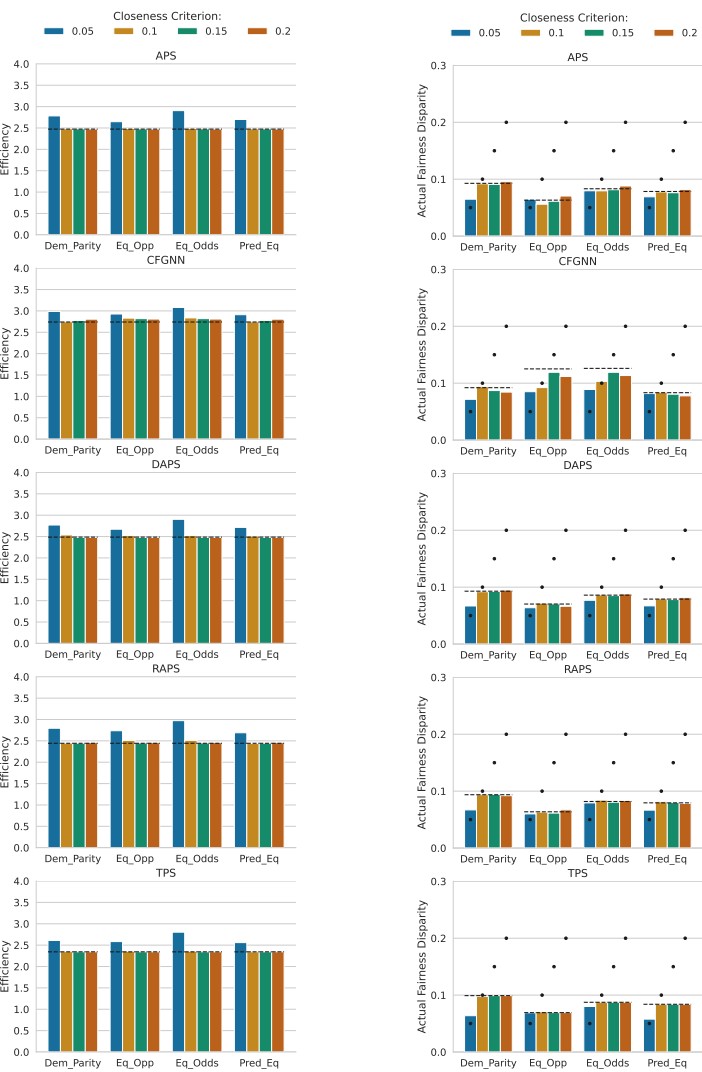

Figure E6: **Pokec-n.** The efficiency (top) and fairness violation (bottom) plots when considering only *gender* as the sensitive attribute. Observe that the baseline disparity here is smaller than the baseline in intersectional fairness (see Figure 3). Thus, when controlling for $c = 0.05$ coverage, there is a minimal change in efficiency across all the non-conformity scores.

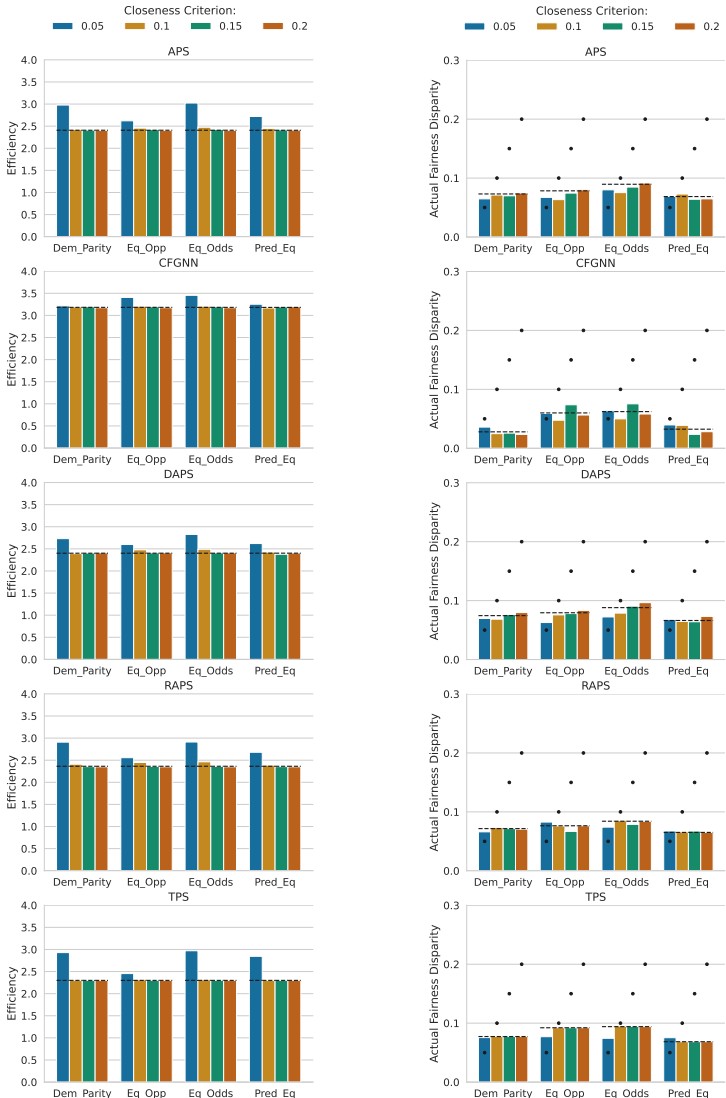

Figure E7: **Pokec-n.** The efficiency (top) and fairness violation (bottom) plots when considering only *region* as the sensitive attribute. Compared to using *gender* in Figure E6, the prediction set sizes are larger for *region* when controlling for $c = 0.05$, thus illustrating that exact performance will vary based on the sensitive attribute and group distribution.

### E.3.2 POKEC-Z

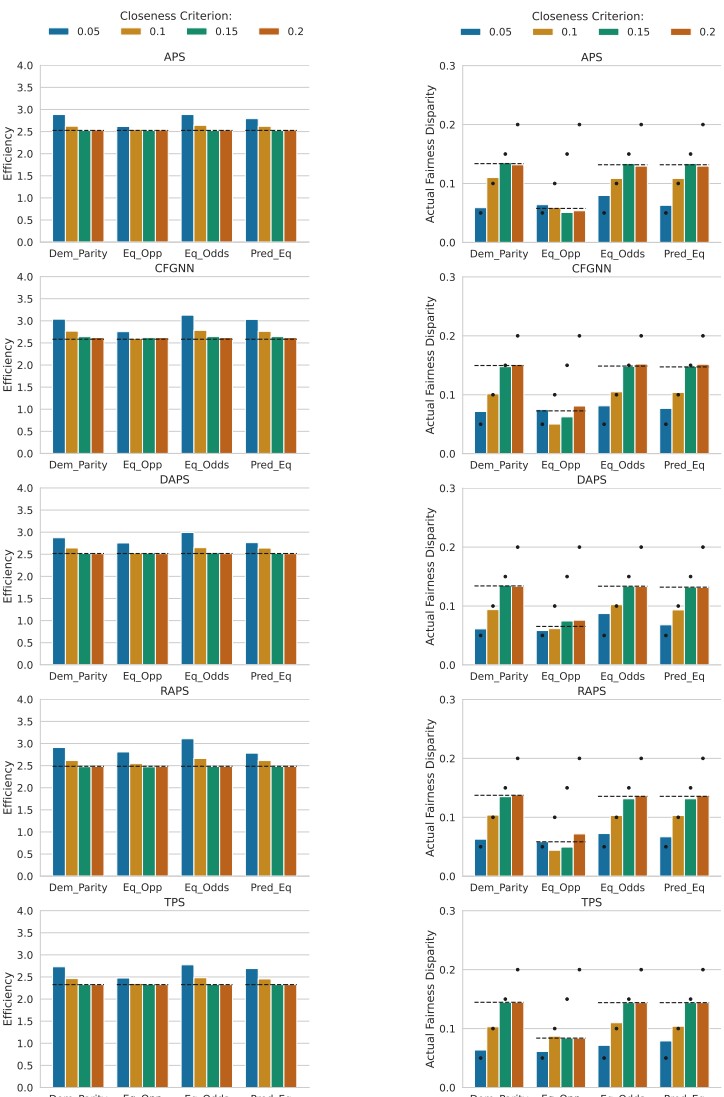

Figure E8: **Pokec-z.** The efficiency (top) and fairness violation (bottom) plots when considering only *gender* as the sensitive attribute. Unlike Pokec-n - using *gender* (see Figure E6 - we find that the baseline is unfair for several values of c - exemplifying the auditing capabilities of the CF framework. This result also demonstrates how fairness can vary at different localities since Pokec-n and Pokec-z are disjoint subgraphs of the same graph.

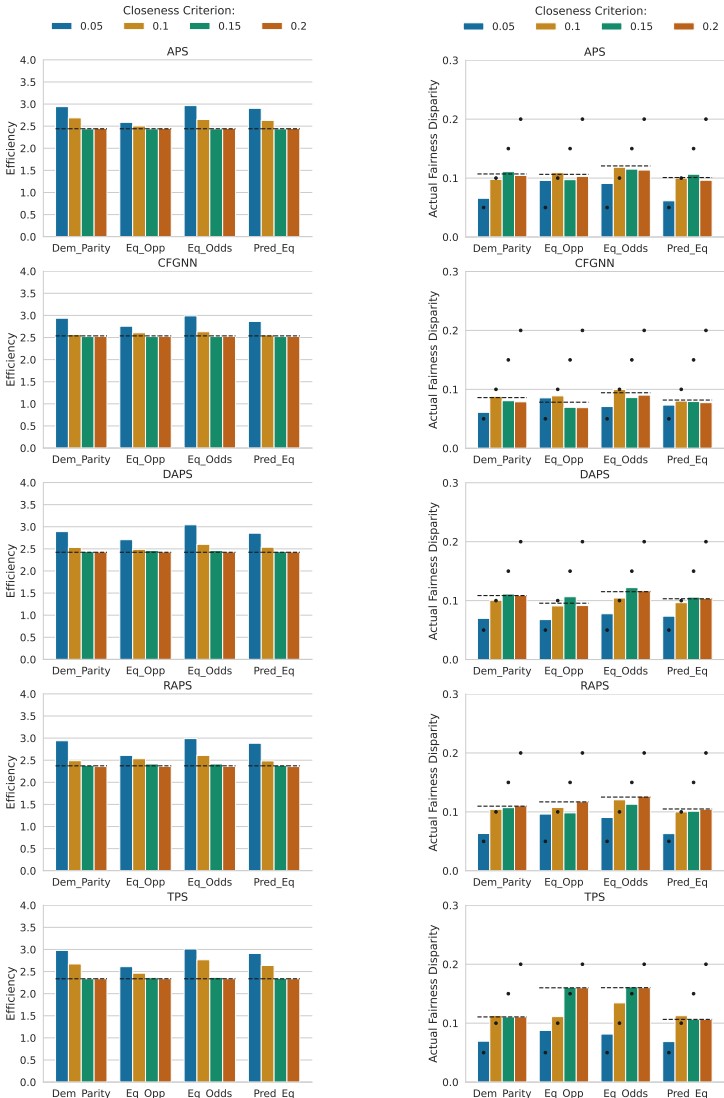

Figure E9: **Pokec-z.** The efficiency (top) and fairness violation (bottom) plots when considering only *region* as the sensitive attribute. In these plots for $c \geq 0.1$, the baselines are fair for all score functions except for TPS. The baseline for TPS being 'unfair' at $c = 0.1$ suggests TPS sacrifices fairness for efficiency.

## E.4 PREDICTIVE PARITY PROXY RESULTS

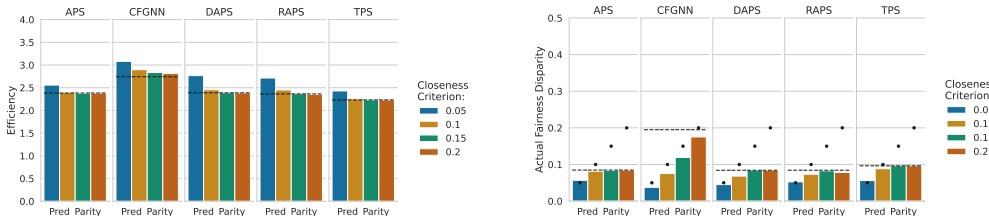

Figure E11: **Pokec-z.** Results using the Predictive Parity Proxy on a *graph* dataset. We can see that the CF Framework controls for this metric as well as or better than the base conformal predictor at all values of $c$ and non-conformity scores.

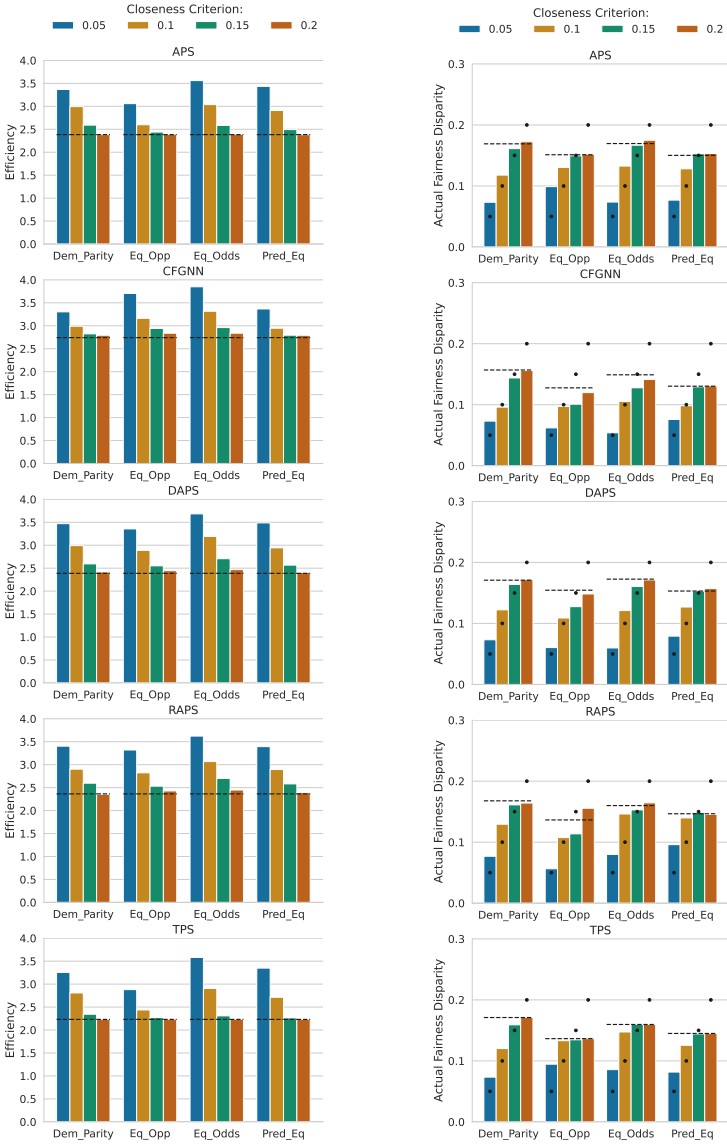

Figure E10: **Pokec-z.** Another example of intersectional fairness using both sensitive attributes of the Pokec-z dataset. Like Pokec-n, we find that the worst-case fairness disparity is better than the baseline for the CF Framework. We reiterate that the challenge of intersection fairness is the multiplicative increase in the number of groups that must be balanced. Thus, intersection fairness would benefit from more calibration points.

## E.5 DISPARATE IMPACT RESULTS

Table E1: **ACSEducation.** The CF framework significantly enhances fairness (increases the ratio close to the desired 80% disparate impact rule), with a cost to efficiency. The classwise approach improves efficiency further while retaining a ratio near 80%.

| | **ACSEducation** | APS | | RAPS | | TPS | |
|---|---|---|---|---|---|---|---|
| Classwise | | Base | CF | Base | CF | Base | CF |
| False | Disp. Impact | 0.011 | **0.799** | 0.014 | **0.791** | 0.019 | **0.804** |
| | Efficiency | 2.982 | 5.662 | 3.031 | 5.658 | 2.828 | 5.705 |
| True | Disp. Impact | 0.011 | **0.781** | 0.014 | **0.761** | 0.019 | **0.797** |
| | Efficiency | 2.982 | 4.727 | 3.031 | 4.724 | 2.828 | 4.744 |

Table E2: **ACSIncome.** The CF framework significantly enhances fairness (increases the ratio close to the desired 80% disparate impact rule), with a cost to efficiency. The classwise approach improves efficiency further while retaining a ratio near 80%.

| | **ACSIncome** | APS | | RAPS | | TPS | |
|---|---|---|---|---|---|---|---|
| Classwise | | Base | CF | Base | CF | Base | CF |
| False | Disp. Impact | 0.397 | **0.815** | 0.387 | **0.847** | 0.356 | **0.804** |
| | Efficiency | 2.212 | 3.557 | 2.169 | 3.610 | 2.109 | 3.416 |
| True | Disp. Impact | 0.397 | **0.797** | 0.387 | **0.790** | 0.356 | **0.797** |
| | Efficiency | 2.212 | 2.674 | 2.169 | 2.752 | 2.109 | 2.679 |

Table E3: **Credit.** For the Credit dataset, we notice that the CF Framework improves over the baseline for APS, RAPS, TPS, and DAPS. We find that the baseline for TPS performs the worst of the 4 methods regarding disparate impact. Interestingly, the CFGNN baseline, on the other hand, maximizes the disparate impact while having the best efficiency. CFGNN demonstrates a case where CF does not perform worse than the baseline since the baseline was already 'fair'. It also provides an example of where an audit via CF would find that the CFGNN conformal predictor is *a priori* fair.

| | **Credit** | APS | | RAPS | | TPS | | CFGNN | | DAPS | |
|---|---|---|---|---|---|---|---|---|---|---|---|
| Classwise | | Base | CF | Base | CF | Base | CF | Base | CF | Base | CF |
| False | Disp. Impact | 0.646 | **0.821** | 0.586 | **0.768** | 0.252 | **0.793** | **0.922** | **0.922** | 0.539 | **0.809** |
| | Efficiency | 2.325 | 2.561 | 2.326 | 2.559 | 2.268 | 2.620 | 2.202 | 2.202 | 2.254 | 2.573 |
| True | Disp. Impact | 0.646 | **0.821** | 0.586 | **0.768** | 0.252 | **0.793** | **0.922** | **0.922** | 0.539 | **0.809** |
| | Efficiency | 2.325 | 2.513 | 2.326 | 2.509 | 2.268 | 2.558 | 2.202 | 2.202 | 2.254 | 2.526 |

Table E4: **Pokec-n.** For Pokec-n, the CF Framework improves the disparate impact of the baseline conformal predictor. Similar to the other fairness metrics, the disparate impact worsens when considering intersectional fairness and is near or exceeds 80% when only one sensitive attribute is considered (i.e., *region* or *gender*).

| Classwise | Group | **Pokec-n** | APS Base | APS CF | RAPS Base | RAPS CF | TPS Base | TPS CF | CFGNN Base | CFGNN CF | DAPS Base | DAPS CF |
|---|---|---|---|---|---|---|---|---|---|---|---|---|
| False | Gender | Disp. Impact | 0.798 | **0.802** | 0.793 | **0.811** | 0.777 | **0.796** | 0.784 | **0.797** | 0.800 | **0.806** |
| | | Efficiency | 2.465 | 2.565 | 2.434 | 2.648 | 2.343 | 2.494 | 2.636 | 2.838 | 2.474 | 2.639 |
| True | Gender | Disp. Impact | 0.798 | **0.802** | 0.793 | **0.810** | 0.777 | **0.796** | 0.784 | **0.797** | 0.800 | **0.806** |
| | | Efficiency | 2.465 | 2.473 | 2.434 | 2.457 | 2.343 | 2.358 | 2.636 | 2.666 | 2.474 | 2.494 |
| False | Region | Disp. Impact | 0.814 | **0.823** | 0.820 | **0.829** | 0.803 | **0.814** | 0.916 | **0.918** | 0.822 | **0.831** |
| | | Efficiency | 2.401 | 2.498 | 2.373 | 2.552 | 2.300 | 2.426 | 3.168 | 3.185 | 2.413 | 2.584 |
| True | Region | Disp. Impact | **0.814** | **0.814** | 0.820 | **0.822** | 0.803 | **0.806** | 0.916 | **0.917** | 0.822 | **0.824** |
| | | Efficiency | 2.401 | 2.430 | 2.374 | 2.404 | 2.300 | 2.332 | 3.168 | 3.180 | 2.413 | 2.437 |
| False | Region & Gender | Disp. Impact | 0.718 | **0.792** | 0.723 | **0.774** | 0.716 | **0.785** | 0.602 | **0.812** | 0.720 | **0.787** |
| | | Efficiency | 2.485 | 3.037 | 2.435 | 3.012 | 2.341 | 2.970 | 2.537 | 3.563 | 2.509 | 2.991 |
| True | Region & Gender | Disp. Impact | 0.718 | **0.767** | 0.723 | **0.760** | 0.716 | **0.754** | 0.602 | **0.779** | 0.720 | **0.764** |
| | | Efficiency | 2.485 | 2.739 | 2.435 | 2.656 | 2.341 | 2.566 | 2.537 | 2.964 | 2.509 | 2.709 |

Table E5: **Pokec-z.** For Pokec-z, the CF Framework improves the disparate impact of the baseline conformal predictor. Similar to the other fairness metrics, the disparate impact worsens when considering intersectional fairness and is near or exceeds 80% when only one sensitive attribute is considered (i.e., *region* or *gender*).

| Classwise | Group | **Pokec-z** | APS Base | APS CF | RAPS Base | RAPS CF | TPS Base | TPS CF | CFGNN Base | CFGNN CF | DAPS Base | DAPS CF |
|---|---|---|---|---|---|---|---|---|---|---|---|---|
| False | Gender | Disp. Impact | 0.798 | **0.802** | 0.737 | **0.800** | 0.737 | **0.800** | 0.784 | **0.797** | 0.800 | **0.806** |
| | | Efficiency | 2.523 | 2.995 | 2.489 | 3.088 | 2.327 | 3.134 | 2.512 | 3.388 | 2.522 | 2.996 |
| True | Gender | Disp. Impact | 0.798 | **0.802** | 0.737 | **0.800** | 0.661 | **0.742** | 0.784 | **0.797** | 0.800 | **0.806** |
| | | Efficiency | 2.523 | 2.572 | 2.489 | 2.571 | 2.327 | 2.415 | 2.512 | 2.642 | 2.522 | 2.579 |
| False | Region | Disp. Impact | 0.807 | **0.823** | 0.796 | **0.826** | 0.812 | **0.816** | 0.781 | **0.796** | 0.811 | **0.829** |
| | | Efficiency | 2.429 | 2.569 | 2.369 | 2.592 | 2.337 | 2.497 | 2.546 | 2.644 | 2.416 | 2.586 |
| True | Region | Disp. Impact | 0.807 | **0.815** | 0.796 | **0.800** | 0.812 | **0.816** | 0.781 | **0.796** | 0.811 | **0.813** |
| | | Efficiency | 2.429 | 2.475 | 2.369 | 2.403 | 2.337 | 2.391 | 2.546 | 2.603 | 2.416 | 2.453 |
| False | Region & Gender | Disp. Impact | 0.658 | **0.787** | 0.661 | **0.770** | 0.640 | **0.785** | 0.590 | **0.806** | 0.658 | **0.768** |
| | | Efficiency | 2.408 | 3.173 | 2.359 | 3.214 | 2.265 | 3.175 | 2.512 | 3.471 | 2.409 | 3.133 |
| True | Region & Gender | Disp. Impact | 0.658 | **0.773** | 0.661 | **0.742** | 0.640 | **0.745** | 0.590 | **0.800** | 0.658 | **0.749** |
| | | Efficiency | 2.408 | 2.799 | 2.359 | 2.741 | 2.265 | 2.627 | 2.512 | 2.788 | 2.409 | 2.743 |

### E.6 COMPARISON WITH BATCHGCP

To produce conformal prediction sets with group-wise coverage, BatchGCP (Jung et al., 2023) learns a group-dependent threshold function to provide $1 - \alpha$ coverage for the correct label - i.e. $\Pr\Big(y_{n+1} \in \mathcal{C}_{\hat{f}(\boldsymbol{x}_{n+1};\lambda)}(\boldsymbol{x}_{n+1}) \mid \boldsymbol{x}_{n+1} \in g'\Big) = 1 - \alpha \ \forall g' \in \mathcal{G}$. To achieve this, a group-dependent threshold function, $\hat{f}(\boldsymbol{x}_{n+1};\lambda)$, in Equation 9 is used to construct prediction sets by adding a correction to the base threshold function $f(x)$. For the scoring functions considered (i.e., APS), we define $f(x) \equiv \hat{q}(\alpha)$ as the $1 - \alpha$ quantile of the calibration scores. $\lambda \in \mathbb{R}^{|\mathcal{G}|}$ is a vector with $\lambda_{g'}$ corresponding to the entry for the $g'$ group. The groups, $g'$, may intersect, allowing for $\boldsymbol{x}_{n+1}$ to be a part of multiple groups.

$$\hat{f}(x;\lambda) = f(x) + \sum_{g' \in G} \lambda_{g'} \mathbf{1}[x \in g'] \tag{9}$$

To determine the value of $\lambda$, Jung et al. (2023) solve the convex optimization problem in 10, where $L_q$ is the pinball loss (Equation 11) - a function used to determine how close a threshold, $\hat{f}(x;\lambda)$, is to a specific quantile, $1 - \alpha$, for a given score, $s$. Using the pinball loss, $\lambda^*$ - the optimal $\lambda$ - is computed and used to construct prediction sets.

$$\lambda^* = \underset{\lambda}{\operatorname{argmin}} \ \mathbb{E}\big[L_q\big(\hat{f}(x;\lambda), s(x,y)\big)\big] \tag{10}$$

$$L_{1-\alpha}(\tau, s) = (s - \tau)(1 - \alpha)\mathbf{1}[s > \tau] + (\tau - s)\alpha\mathbf{1}[s \leq \tau] \tag{11}$$

To compare against BatchGCP, we adapted the codebase[3] provided by Jung et al. (2023) to accommodate APS as a non-conformity score and use the classification variant of the Folktables datasets since Jung et al. (2023) conducts experiments with those datasets. With the BatchGCP implementation, we get the threshold function, $\hat{f}$, and use it when constructing prediction sets. The fairness disparity is evaluated using Demographic Parity and Disparate Impact. Since BatchGCP aims to optimize *group*-conditional coverage, we only compare against those two fairness metrics, for a fair comparison.

While BatchGCP provides PAC guarantees on group-wise coverage, it does not *necessarily* provide the fairness guarantees our framework does, as empirically seen with the ACSIncome and ACSEducation datasets in Table E6 and Figures E12 and E13. We can dissect the poor performance on these datasets by looking at the per-group conditional coverages in Figure E14, where several groups are undercovered. We note that BatchGCP *requires group information at inference time*, which restricts it from settings where group information may be unavailable at inference time – this is not a limitation of the CF Framework.

Table E6: Comparing Base APS, BatchGCP, and CF framework under Disparate Impact. For the CF framework, we set $c = 0.8$ and use classwise lambdas. We observe that the CF Framework can achieve a disparate impact value much closer to $c = 0.8$ with little effect on efficiency.

|  |  | Base APS | BatchGCP | CF |
|---|---|---|---|---|
| **ACSEducation** | Disp. Impact | 0.011 | 0.576 | **0.781** |
|  | Efficiency | 2.982 | 2.893 | 4.727 |
| **ACSIncome** | Disp. Impact | 0.397 | 0.349 | **0.797** |
|  | Efficiency | 2.212 | 2.200 | 2.674 |

---

[3] https://github.com/ProgBelarus/BatchMultivalidConformal

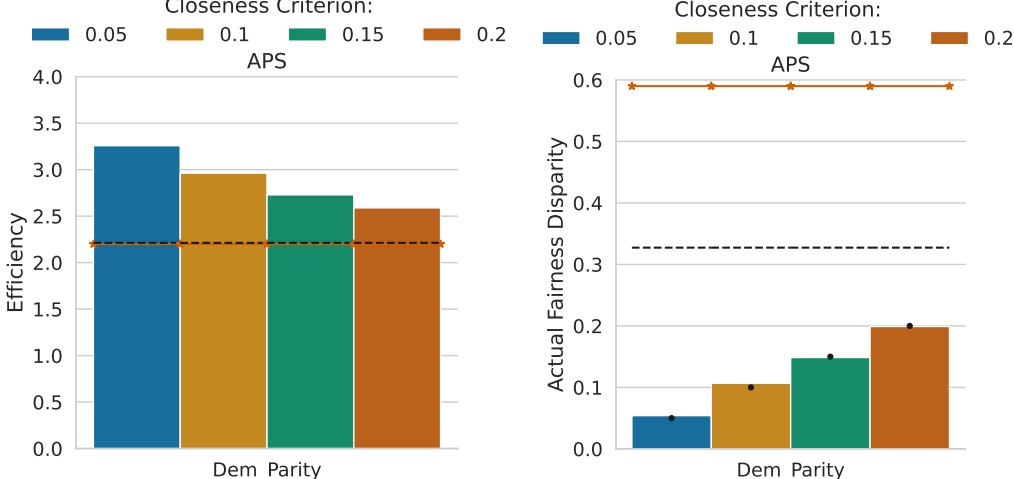

Figure E12: **ACSIncome.** Comparing Base APS, BatchGCP, and CF framework under Demographic Parity. For the CF framework, we vary $c \in \{0.05, 0.1, 0.15, 0.2\}$. We observe that the CF Framework achieves the smallest disparity for every value of $c$ (seen on the right figure) with a small cost to the efficiency (as seen on the left figure). In the figure, Base APS = Black lines, BatchGCP = Red lines, and the CF framework is the bar charts. The black dots are the *desired* disparity level for the CF framework, which the CF framework achieves, while neither Base APS nor BatchGCP meets these thresholds.

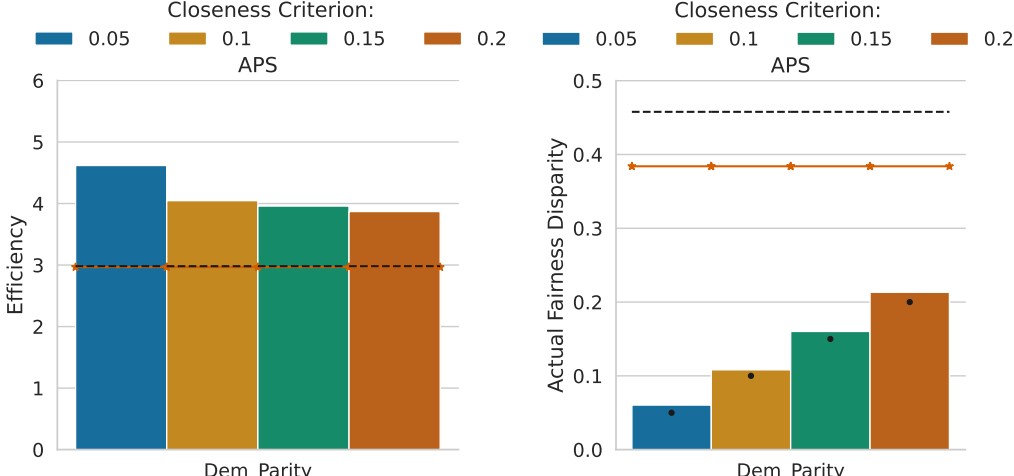

Figure E13: **ACSEducation.** Comparing Base APS, BatchGCP, and CF framework under Demographic Parity. For the CF framework, we vary $c \in \{0.05, 0.1, 0.15, 0.2\}$. We observe that the CF Framework achieves the smallest disparity for every value of $c$ (seen on the right figure) with a small cost to the efficiency (as seen on the left figure). In the figure, Base APS = Black lines, BatchGCP = Red lines, and the CF framework is the bar charts. The black dots are the *desired* disparity level for the CF framework, which the CF framework achieves, while neither Base APS nor BatchGCP meets these thresholds.

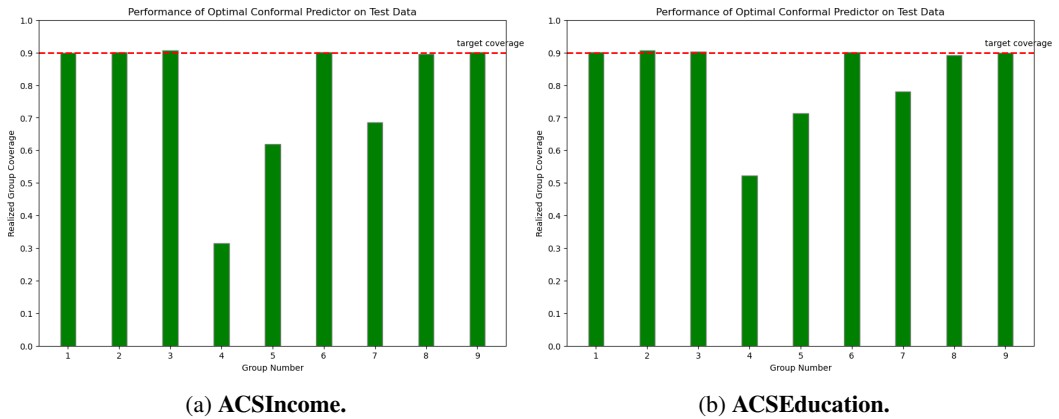

(a) **ACSIncome.**                    (b) **ACSEducation.**

Figure E14: To dissect the poor performance of BatchGCP seen in Figures E12 and E13, we present the per-group conditional coverages for both datasets and see that certain groups are significantly undercovered (i.e., groups 4, 5, and 7 in both figures).

