# OpenReview forum: "A Generic Framework for Conformal Fairness"
_ICLR.cc/2025/Conference — ICLR 2025 Poster_

### Official Review · Reviewer_3NSU · 2024-10-27

**Soundness:** 3
**Presentation:** 3
**Contribution:** 2
**Rating:** 6
**Confidence:** 2

**Summary:**

This paper proposes a notion of fairness, named $\textit{conformal fairness}$. In contrast with standard fairness notions (e.g. demographic parity), this notion significantly applies conformal mapping, which the authors clearly defined. The authors developed an algorithm to achieve this type of fairness on graph datas and confirmed with experiments.

**Strengths:**

1. This paper is well-written with a clear structure.

2. The notion of conformal prediction is common, while its application on fairness is not widely studied to the best of my knowledge. The novelty of this notion is well justified.

3. The conformal fairness metrics in Table 1 are intuitive and well-founded.

**Weaknesses:**

1. Although the conformal fairness metrics are well-defined in Table 1, they were not carefully discussed in later results (Theorem 3.4). Specifically, Theorem 3.4 states that "the coverage difference for Predictive Parity is satisfied". In Table 1, the definition of Predictive Parity relies on a conformal prediction $\mathcal{C}$, but the specific mapping/quantification is not discussed in the theorem statement.

2. This is a continuation of the previous point. Suppose there is a conformal prediction in Theorem 3.4, how good is it, i.e. what shall the value of $\alpha$ be, as in Equation 1?

**Questions:**

Please see the weaknesses section.

---

> ### Author Response · Authors · 2024-11-20
> **Response to reviewer**
>
> We thank the reviewer for taking the time to review our paper. We further appreciate the reviewer’s positive comments regarding the manuscript's writing.
>
> > [W1] Conformal Fairness metrics need to be well-discussed in methodology. Specifically, Theorem 3.4 states that "the coverage difference for Predictive Parity is satisfied". In Table 1, the definition of Predictive Parity relies on a conformal prediction, $\mathcal{C}$, but the specific mapping/quantification is not discussed in the theorem statement
>
> We are happy to clarify Theorem 3.4’s statement, especially in the last sentence regarding “the coverage difference for Predictive Parity is satisfied”. The use of “coverage difference” is a bit misleading since the prediction set is the conditional of the probability for Predictive Parity. We are balancing an adapted PPV between groups by minimizing the difference. We will clean up the language in Theorem 3.4 in the updated manuscript.
>
> > [W2] Suppose there is a conformal prediction in Theorem 3.4, how good is it, i.e. what shall the value of $\alpha$ be, as in Equation 1
>
> The choice of $\alpha$ may not be in the hands of the practitioner but instead defined by regulatory frameworks within the domain you’re focused on (e.g. 80% rule is $\alpha = 0.8$ for Disparate Impact). This is similar to standard conformal prediction, where you have an $\alpha$ miscoverage that you want to be within.

---

> > ### Comment · Reviewer_3NSU · 2024-11-20
> > **Response to authors**
> >
> > Thank you for clarifying my questions. I will keep my score.

---

> > > ### Author Response · Authors · 2024-11-28
> > > **Thank You**
> > >
> > > We thank the reviewer for engaging in a discussion about our work. We are happy to answer any further questions the reviewer may have.

---

### Official Review · Reviewer_VVws · 2024-11-02

**Soundness:** 3
**Presentation:** 2
**Contribution:** 3
**Rating:** 6
**Confidence:** 2

**Summary:**

This paper proposes a framework to check and find the suitable threshold for conformal learning. Further, the framework can be easily extended to some constraints, like fairness constraints. The authors specifically studied achieving fairness properties for graph models. Further, the authors studied how to use their framework to check the fairness property of a model. The authors' algorithm first find the right use inverse quantile to check whether a threshold is good.

**Strengths:**

Conformal prediction is a very interesting area and there are many interesting works in this area. It is a natural question to ask how to achieve fairness for models in this setting. The authors provide a very general framework that can achieve many different definitions of algorithmic fairness. The paper also did many interesting experiments in graph datasets.

**Weaknesses:**

I personally think the author does not elaborate enough on why they chose this algorithm design. The writing of this paper could be improved.

**Questions:**

I think it would be very helpful for the authors to explain more about their design choices. For example, why choose inverse quantile for miscoverage level? Explain a bit more about the intuitions behind lemma 3.1 and 3.2 would really help the readers.

---

> ### Author Response · Authors · 2024-11-20
> **Response to reviewer**
>
> We thank the reviewer for taking the time to review our paper.
>
> > [W1] Need to elaborate on design choices.\
> > [Q1] Why choose inverse quantile for miscoverage level? Explain a bit more about the intuitions behind Lemmas 3.1 and 3.2.
>
> We are happy to elaborate on our design choices in the methodology section. Below, we’ve provided some motivation for the particular queries the reviewer asked about.
>
> **Choice of Inverse Quantile:** To achieve a $1 - \alpha$ coverage level (or equivalently an $\alpha$ miscoverage level), prediction sets are constructed using the $(1 - \alpha)$ quantile of the calibration scores as an upper threshold for when to include class labels. Formally, $\mathbb{P}(y_{n + 1} \in \mathcal{C}(x_{n + 1})) = \mathbb{P}(s(x_{n + 1}, y_{n + 1})\leq \hat{q}(\alpha)) = Q^{-1}(\hat{q}(\alpha), \lbrace s(x_i, y_i)\rbrace_{i = 1,}^{n})$, where $Q^{-1}$ is the inverse quantile function. Varying the miscoverage can be done by varying the threshold used to be higher or lower than $\hat{q}(\alpha)$.
>
> **Lemma 3.1 Motivation and Intuition:** The fairness metrics considered in this work are evaluated on a subpopulation (e.g. for members of a particular group or particular true label). The purpose of Lemma 3.1 is to assert the bounds in CP hold when we consider a subset of the calibration set, which intuitively makes sense due to exchangeability.
>
> **Lemma 3.2 Motivation and Intuition:** Lemma 3.2 is to derive the miscoverage level when given a particular threshold using the calibration scores. This workflow is the opposite of conformal prediction, where you are given a miscoverage level and need to determine the threshold (i.e. $\hat{q}(\alpha)$) that achieves it. The intuition for the inequality comes from exchangeability between the calibration and test scores.

---

> > ### Comment · Reviewer_VVws · 2024-11-23
> >
> > Thank you for answering my questions. I will keep my score.

---

> ### Author Response · Authors · 2024-11-28
> **Thank You**
>
> We thank the reviewer for engaging in a discussion about our work. We are happy to answer any further questions the reviewer may have.

---

### Official Review · Reviewer_zT1T · 2024-11-03

**Soundness:** 3
**Presentation:** 2
**Contribution:** 3
**Rating:** 6
**Confidence:** 3

**Summary:**

The paper introduces "Conformal Fairness," a framework that extends Conformal Prediction (CP) to ensure fairness across sensitive groups. The authors develop a theoretically grounded algorithm to control gaps in coverage between these groups, leveraging the exchangeability assumption instead of the typical IID assumption. This allows the framework to be applied to non-IID data types, including graph data. Experiments on both graph and tabular datasets show that the framework can effectively control fairness gaps while maintaining coverage guarantees.

**Strengths:**

- Originality: Introduces "Conformal Fairness," extending Conformal Prediction (CP) to address fairness in uncertainty quantification, especially for non-IID data.
- Quality: Provides strong theoretical backing with rigorous proofs and effective validation through experiments on graph and tabular datasets.
- Clarity: Clearly defined theoretical concepts and a stepwise algorithm description make the methodology accessible to those with relevant background knowledge. Providing additional background information on key concepts would help readers who are less familiar with the topics to follow more easily
- Significance: Tackles an important gap by combining fairness with uncertainty quantification. Adaptable to multiple fairness metrics and data types, making it broadly applicable in ethical AI contexts.

**Weaknesses:**

- The paper lacks a detailed discussion of the fairness-efficiency trade-off. Quantifying acceptable efficiency losses when fairness is improved would make the results more actionable for practitioners balancing both aspects.

- The extension of the exchangeability assumption to real-world data may not always hold. Adding empirical evidence or discussion on when this assumption is valid in practice would make the claims more robust.

-Lack of comparison with all existing fairness-aware methods limits benchmarking. Adding baselines would clarify the framework's effectiveness.

**Questions:**

- How does the framework handle scalability with multiple sensitive attributes, especially computationally?

- Why were existing fairness-aware methods not used as baselines for comparison?

- How practical is the exchangeability assumption for real-world graph data? Any empirical evidence?

---

> ### Author Response · Authors · 2024-11-20
> **Response to reviewer [1/2]**
>
> We thank the reviewer for taking the time to review our paper.
>
> > [W1] Lacking detailed discussion of the fairness-efficiency trade-off. Quantifying acceptable efficiency losses when fairness is improved would make the results more actionable for practitioners balancing both aspects
>
> Figures 1 and 2 in the main paper depict the fairness-efficiency trade-off at a high level, but as is common when discussing utility-fairness relationships in the literature [1, 2], improving fairness comes at the cost of utility (i.e., efficiency). The particular requirements for fairness and efficiency losses may not be in the hands of the practitioner but set by regulatory frameworks that exist in the domain you’re focused on (e.g. 80% Rule for Disparate Impact).
>
> > [Q1] How does the framework handle scalability with multiple sensitive attributes, especially computationally?
>
> Currently, the framework is implemented serially and thus scales with the number of sensitive groups; however, the framework is embarrassingly parallel with respect to the $\Lambda$ search space (Line 2 in Algorithm 1) and with the inverse quantile computations for each (sensitive group, label) pair (Lines 9-14 in Algorithm 1). Thus, distributed frameworks (e.g., Ray) can be integrated to utilize multiple cores. Parallelizing Lines 9-14 can alleviate the effect of the number of sensitive attributes for framework scalability.
>
> > [W2] The extension of the exchangeability assumption to real-world data may not always hold. Adding empirical evidence or discussion on when this assumption is valid in practice would make the claims more robust\
> > [Q3] How practical is the exchangeability assumption for real-world graph data? Any empirical evidence?
>
> In this work, we consider the transductive setting for node classification which, in the existing conformal prediction on graphs literature, there is a theorem that states that the exchangeability assumptions hold between the calibration and test nodes in the transductive setting [3, 4]. We don’t make any claims about the inductive setting, where additional assumptions are required and exchangeability may not be achieved.\
> \
> \
> [1] Yingqiang Ge, Xiaoting Zhao, Lucia Yu, Saurabh Paul, Diane Hu, Chu-Cheng Hsieh, and Yongfeng Zhang. 2022. Toward Pareto Efficient Fairness-Utility Trade-off in Recommendation through Reinforcement Learning. In Proceedings of the Fifteenth ACM International Conference on Web Search and Data Mining (WSDM '22).
>
> [2] S. Dehdashtian, B. Sadeghi and V. N. Boddeti, "Utility-Fairness Trade-Offs and how to Find Them," 2024 IEEE/CVF Conference on Computer Vision and Pattern Recognition (CVPR)
>
> [3] Huang, Kexin, Ying Jin, Emmanuel Candes, and Jure Leskovec. "Uncertainty quantification over graph with conformalized graph neural networks." Advances in Neural Information Processing Systems 36 (2024).
>
> [4] Soroush H. Zargarbashi, Simone Antonelli, and Aleksandar Bojchevski. Conformal prediction sets for graph neural networks. In Andreas Krause, Emma Brunskill, Kyunghyun Cho, Barbara Engelhardt, Sivan Sabato, and Jonathan Scarlett, editors, Proceedings of the 40th International Conference on Machine Learning, volume 202 of Proceedings of Machine Learning Research, pages 12292–12318. PMLR, 23–29 Jul 2023.

---

> ### Author Response · Authors · 2024-11-20
> **Response to reviewer [2/2]**
>
> > [W3] Lack of comparison with all existing fairness-aware methods limits benchmarking. Adding baselines would clarify the framework's effectiveness.\
> > [Q2] Why were existing fairness-aware methods not used as baselines for comparison?
>
> Our paper is focused on making a fair version of conformal prediction, thus we compare it to standard conformal prediction methodologies (e.g., APS and TPS). Our framework can utilize other CP methods (e.g. RAPS, CF-GNN, DAPS, etc.) and directly improve on those in terms of fairness. For an additional non-conformity score method, we include results with RAPS in Figure E14 in "Rebuttal Additional Results.pdf" in the supplemental material.
>
> Going beyond simply using group conditional coverage directly, which is currently our baseline, there hasn’t been much work in the fairness + conformal prediction space. As mentioned in our Discussion section, existing works at the intersection of fairness and conformal prediction for comparison are limited. Three recent works we mentioned are cited (Wang et al., 2024), (Lu et al., 2022), and (Liu et al., 2022).
>
> Wang et al. introduce the notion of Equal Opportunity of Coverage – a conformal notion of Equal Opportunity. This work is an instantiation of our work specifically for Equal Opportunity. Our work generalizes to several popular fairness and user-defined metrics, like our Predictive Parity Proxy, which could not be considered in Wang et al.
>
> Lu et al. focus on a particular application of skin lesion classification for medical testing. Furthermore, their proposed method – GAPS/GRAPS – requires group information at inference time, which our method does not and is not always viable in online settings.
> Liu et al. focus on fairness for quantile regression and tailor their framework for Demographic Parity. Furthermore, their proposed method requires group information at inference time, which our method does not and is not always viable in online settings.
>
> Addressing Reviewer ST3N’s comments about having another baseline related to the recent group-conditional CP literature, we compare against BatchGCP [5] as a baseline to demonstrate the CF framework’s efficacy in balancing miscoverage across groups for all labels. The comparison is done for Disparate Impact and Demographic Parity as they both only require group conditioning. The results are presented in Table E6 and Figure E12 in "Rebuttal Additional Results.pdf" in the supplemental material.
>
> We plan to include these additional results in an appendix in the updated manuscript.\
> \
> \
> [5] C. Jung et al: Batch Multivalid Conformal Prediction [ICLR 2023]

---

> > ### Comment · Reviewer_zT1T · 2024-11-23
> >
> > Thank you for clarifying my questions. I feel your paper can add value to the conference, so I will maintain my score

---

> > > ### Author Response · Authors · 2024-11-28
> > > **Thank You**
> > >
> > > We thank the reviewer for engaging in a discussion about our work and acknowledging its value for conference attendees. We are happy to answer any further questions the reviewer may have.

---

### Official Review · Reviewer_ST3N · 2024-11-04

**Soundness:** 3
**Presentation:** 2
**Contribution:** 3
**Rating:** 6
**Confidence:** 4

**Summary:**

The manuscript introduces and studies a general-purpose conformal fairness framework. It consists of an algorithm that can enforce various notions of fairness (such as demographic parity, equalized odds, predictive parity, and several others contained in the ML fairness literature) on the coverage of prediction sets, in the sense of satisfying appropriate conformal guarantees on expected coverage conditional on sensitive labels/attributes. To demonstrate practical performance of the framework given the various fairness notions, experimental evaluation is provided on a variety of supervised learning tasks; besides the standard setting, graph conformal prediction with fairness constraints is also demonstrated in experiments.

**Strengths:**

The paper's main asset is a reasonably broad evaluation of the proposed methods on a variety of datasets where fairness concerns may necessitate using one of the tabulated conformal fairness metrics. Compared to prior works the experimental section appears more extensive, both in terms of metrics and datasets. Moreover, conformal fairness guarantees were displayed on graph data in addition to the standard supervised tasks. The results indicate that explicitly enforcing fairness according to any of the tabulated fairness criteria is in fact often necessary, as evidenced by vanilla methods not satisfying non-enforced fairness coverage constraints. Furthermore, no drastic deterioration in efficiency guarantees is observed, indicating that the price to pay for fairness conditional coverage may often be acceptable.

**Weaknesses:**

--- The main weakness and bottleneck at this point is the writing of the manuscript. In particular, the writing of Section 3, which introduces the objectives, the general algorithm, and the analysis, is currently not acceptable. Indeed, lots of key notions and terms, both on the fairness side and on the conformal side, are not properly and unambiguously introduced and/or are discussed in arbitrary order.
(Fairness: groups/group collections are never formally defined, nor are requirements on them specified (e.g. non-intersectionality etc); "filters" and "filter functions" are simply thrown into the presentation. No background on the fairness measures, no elaboration on how they are transformed into conformal analogues compared to original notions, except for an incoherent sentence starting with "Essentially achieved..." in line 130. Conformal prediction: auxiliary notions like "interval widths", "label miscoverages" appear right away in the pseudocode, which also uses clunky notation and for which the textual explanation is as confusing as the pseudocode notation.) This is not to say that one cannot fill in many of the details with enough imagination and enough expertise, but at the moment the writing is hardly structured and accessible enough.

--- The theoretical/methodological side does not offer strong novel contributions; for the most part it simply wraps class-conditional and group-conditional conformal methodology into fairness-specific terminology --- and is currently doing it in a somewhat confusing way, given the currently unsatisfactory presentation as stated above.

--- In addition, the literature background review, where it's claimed that there are "very few prior efforts" on fairness and conformal prediction, misses an established line of work on group-conditional fairness guarantees; these works study the enforcement of coverage guarantees (usually of both upper and lower bounds) on rich classes of subpopulations given by possibly arbitrarily overlapping groups.

R. Barber et al: The limits of distribution-free conditional predictive inference [Journal of IMA 2020]
C. Jung et al: Batch Multivalid Conformal Prediction [ICLR 2023]
Z. Deng et al: Happymap: A generalized multi-calibration method [ITCS 2023]
O. Bastani et al: Practical adversarial multivalid conformal prediction [NeurIPS 2022]
I. Gibbs et al: Conformal Prediction with Conditional Guarantees [2023]

**Questions:**

The main desideratum is a substantial overhaul of the paper's writing, especially as it pertains to the methodology Section 3.

There are other minor concerns, such as the exact conformal methods that the current proposed method is tested against. For instance, APS has been superseded in the literature by RAPS, so that should be included in the comparison. Also, importantly, at least a couple of the many recently appearing class-conditional and local conformal methods should also be included in the comparison for the sake of fairness; indeed, these methods are designed to be able to empirically attain partial versions of conditional coverage without receiving explicit constraints (such as fairness ones) as input, so they should provide a stronger baseline than methods like APS.

----------------------------------------------------------------------------------

UPDATE: The rebuttal, and the post-rebuttal manuscript update, address the immediate concern of improving the presentation of key terms/concepts in Section 3 and beyond. Other improvements such as updates to the experimental section were also made. Given these, I have raised my overall score to 6.

---

> ### Author Response · Authors · 2024-11-20
> **Response to reviewer [1/4]**
>
> We thank the reviewer for taking the time to review our paper.
>
> > [W1] The main weakness and bottleneck at this point is the writing of the manuscript In particular, the writing of Section 3, which introduces the objectives, the general algorithm, and the analysis. Lots of key notions and terms, both on the fairness side and on the conformal side, are not properly and unambiguously introduced and/or are discussed in arbitrary order.
>
> We understand the reviewer’s concerns about the clarity and organization of the manuscript and agree that the introduction of key notions and terms and the flow of the material can be improved. We clarify the specific notions/terms the reviewer mentioned below:
>
> **Fairness:**
>
> - **Groups:** We define groups as subpopulations of the dataset attributed to some protected attribute (e.g. race, gender).
>
> - **Filters/Filter function:** Each fairness metric is evaluated on a subpopulation based on the group information and true class label. The “filter function” abstracts the different subpopulations in the algorithm so that inverse quantiles and scores are computed on the correct subpopulations. For a concrete example, see Section 3.3. Input.
>
> - **Fairness → Conformal Fairness:** We adapt the fairness definitions for our conformal framework, by replacing predicted label equality ($\cdot = \hat{Y}$) with existence in the prediction set $\cdot \in\mathcal{C}_{\lambda}(X)$. This sort of conversion is also done in the literature [1].
>
> **Conformal Prediction:**
>
> - **Interval Widths:** Our setting, like most CP works, deals with finite-sample miscoverage guarantees, which state that the miscoverage is in the interval $(\alpha - \frac{1}{n + 1}, \alpha]$. The interval width is $\frac{1}{n + 1}$.
>
> - **Label Miscoverages:** Our framework aims to balance the gaps in miscoverages between sensitive groups _for all advantaged/positive labels_ (e.g., is_approved for loan approval task). Thus, we compute the miscoverage for each group and each positive label using the inverse quantile and save the results.

---

> ### Author Response · Authors · 2024-11-20
> **Response to reviewer [2/4]**
>
> > [W2] The theoretical/methodological side does not offer strong novel contributions; for the most part, it simply wraps class-conditional and group-conditional conformal methodology into fairness-specific terminology
>
> We note that novelty is certainly in the eye of the beholder and that while our framework builds upon the ideas of conditional conformal methodologies, the simplicity of the framework does not necessarily take away from the novelty. Additionally, our framework differentiates itself from the existing literature in a few different aspects:
>
> 1. **Group Information at Inference:** Existing works in group conditional prediction require knowledge of the group information at inference time. This can be a limiting factor as the group information can be purposefully withheld to de-bias certain processes or in an online setting, where data is coming in a real-time or streaming manner as mentioned in Lines 238-241. In these settings, even if group information is absent, the CF framework still provides group-conditional coverage. We will elaborate on this point in the updated manuscript as we clean up the writing in response to [W1].
>
> 2. **Balancing miscoverage for all groups and all labels:** Existing literature aims to minimize group conditional miscoverage, i.e., for all groups, $g$, $\mathbb{P}(y_{n + 1}\not\in \mathcal{C}(x_{n + 1})|x_{n + 1}\in g)\leq \alpha$ for some $\alpha$. What our framework does differently than the existing literature, is formulate fairness as the pairwise miscoverage disparity across groups. Additionally, we account for conditioning on advantaged labels, similar to fairness metrics like Equal Opportunity and Predictive Equality, in the disparity measurement, which is essential for evaluating certain notions of fairness. For example, metrics like Equal Opportunity require expressions of the form $\mathbb{P}(y_{n + 1}\not\in \mathcal{C}(x_{n + 1})|x_{n + 1}\in g, \mathbf{y_{n + 1} = y^+})$, which would not fit into the existing group-conditional CP literature.
>
>    To balance the miscoverages between groups, we use the inverse quantile to determine the miscoverage for a given threshold. The inverse quantile operation is unique in this context and is not similarly utilized in the existing literature including the literature the reviewer shared. This operation is performed for all the subpopulations (i.e. based on group membership and true label). Coupled with the bounds given by Lemma 3.1 and 3.2, we can find an appropriate threshold where the fairness disparity requirement and the standard coverage guarantees are met.
>
> 3. **Predictive Parity (Conditioning on prediction set):** Unlike the other fairness metrics considered, Predictive Parity involves conditioning on the predicted label (or existence in the prediction set for conformal fairness). We provide Theorem 3.4, which discusses when the standard notion of Predictive Parity can be balanced. We also present a proxy for Predictive Parity where a solution that satisfies an arbitrarily small disparity constraint (i.e. arbitrarily small values of $c$) always exists.

---

> ### Author Response · Authors · 2024-11-20
> **Response to reviewer [3/4]**
>
> > [W3] Missed an established line of work on group-conditional fairness guarantees; these works study the enforcement of coverage guarantees (usually of both upper and lower bounds) on rich classes of subpopulations given by possibly arbitrarily overlapping groups.
>
> We thank the reviewer for highlighting this line of work and will include these references to better elucidate the developments that lead up to our work.
>
> While these works study group-conditional coverage guarantees, they omit discussions of class conditionality, which is incorporated in the standard (and our conformal) definitions of common fairness metrics. Class-conditional coverage goes beyond group-conditional coverage and is more interpretable than group-conditional coverage in certain settings, as discussed in [2] which we cite in our work.
>
> Below we discuss each of the papers the reviewer shared and how it differs and relates to our work. We pick the only work (C. Jung et al.) among these that can act as a baseline – for the group-conditional-only fairness metrics – and compare our proposed method.
>
> - _**R. Barber et al: The limits of distribution-free conditional predictive inference [IMA 2020]:**_ \
>   This work provides a thorough theoretical analysis of conditional conformal prediction and characterizations for when conditional coverage guarantees hold. Our work with the adapted conformal fairness metrics is an instantiation of the characterizations of the theoretical results presented.
>
> - _**C. Jung et al: Batch Multivalid Conformal Prediction [ICLR 2023], O. Bastani et al: Practical adversarial multivalid conformal prediction [NeurIPS 2022]**:_\
>   These works aim to achieve multivalid conformal guarantees which require (1) group-conditional and (2) threshold-calibrated coverage guarantees. Though our work and these two works involve group-conditional coverage, these works are orthogonal because they focus on additionally achieving threshold-calibrated coverage whereas we consider additionally achieving class-conditional coverage for metrics like Equal Opportunity and Predictive Equality. These methods do not provide applicable guarantees for balancing the coverage of each positive label between groups.\
>   \
>   O. Bastani et al. conducted experiments for sequential prediction tasks – a distinct task from what we are considering.\
>   \
>   C. Jung et al. only conducted experiments for regression-based tasks; however, we were able to use their BatchGCP algorithm (and code) to provide another baseline to compare our method. Using the ACSIncome dataset and the XGBoost model that we trained for our results, we use their BatchGCP algorithm to construct a threshold function and evaluate the fairness disparity and efficiencies compared to our group-conditional baseline and our CF Framework. Since BatchGCP only optimizes for group-conditional coverage, we present the results for Demographic Parity (see Figure E12 in "Rebuttal Additonal Results.pdf" in the supplemental material) and Disparate Impact (see Table E6 in "Rebuttal Additonal Results.pdf" in the supplemental material) – two fairness metrics that only condition on group information. The results show that our framework can control for the group conditional coverage disparity between groups unlike Base APS and BatchGCP. To dissect the poor performance of BatchGCP, we plot the per-group coverages in Figure E13 in "Rebuttal Additonal Results.pdf" in the supplemental material and observe that certain groups are significantly undercovered (i.e., groups 4, 5, and 7).
>
> - _**Z. Deng et al: Happymap: A generalized multi-calibration method [ITCS 2023]:**_\
>   This work introduces a generalization for multi-calibration and how it relates to algorithmic fairness for conformal prediction; however, the applications are limited to Equalized Coverage for Regression tasks and don’t present any experiments (synthetic or real) to compare against.
>
> - _**I. Gibbs et al: Conformal Prediction with Conditional Guarantees [2023]:**_\
>   This work considers a wide array of problems that interpolate between marginal and conditional coverage by considering conditional coverage as a class of covariate shifts. This work generalizes some of the notions presented in C. Jung et al. regarding group conditional coverages; however, it relies on the i.i.d. assumption throughout its theory. This assumption limits the work’s applicability to non-i.i.d spaces (e.g. graphs) which our framework applies as seen with our experiments on real-world graph datasets.
>
> [1] Anastasios Nikolas Angelopoulos, Stephen Bates, Adam Fisch, Lihua Lei, Tal Schuster, “Conformal Risk Control”, The Twelfth International Conference on Learning Representations (2024)
>
> [2] Tiffany Ding, Anastasios N. Angelopoulos, Stephen Bates, Michael I. Jordan, and Ryan J. Tibshirani. Class-conditional conformal prediction with many classes. In Proceedings of the 37th International Conference on Neural Information Processing Systems, NIPS ’23

---

> ### Author Response · Authors · 2024-11-20
> **Response to reviewer [4/4]**
>
> > [Q1] Minor Concerns: RAPS has superseded APS. Also, need some baselines with other conditional conformal prediction methods.
>
> The CF Framework is agnostic to the particular non-conformity scoring function and can easily work with RAPS as it does APS, TPS, CF-GNN, DAPS, etc. In Figure E14 in "Rebuttal Additional Results.pdf" in the supplemental material, we show some results for efficiency and fairness violations for varying values of $c$ with the ACSIncome dataset with RAPS alongside the APS results from the manuscript. We observe that the CF framework improves the fairness compared to the baseline RAPS predictor similarly to the APS results.
>
> For comparison with other conditional CP methods, see the results shared for [W3] where we adapt the BatchGCP algorithm for the ACS datasets.
>
> We plan to include these additional results in an appendix in the updated manuscript.

---

> > ### Comment · Reviewer_ST3N · 2024-11-25
> >
> > Thank you for the detailed rebuttal. I have carefully read it, alongside the other reviews and the similarly detailed rebuttals to them and to all reviewers. I have also read the updated manuscript and appendix. In my opinion, these updates and the rebuttals have improved the submission, in particular addressing issues pointed out in my review, and I am correspondingly raising my score to "weak accept" to reflect that. Furthermore, I have no major followup questions to the authors.
> >
> > My main concern about the submission, as I stated in the review above, was the less-than-satisfactory exposition in Section 3, and I am happy to report that the updates to the manuscript have substantially addressed this concern. In particular, updates such as the filter function notation (now both well-defined in the text and propagated into the algorithm) should make considerably more clear to the reader the ways in which the proposed framework is able to generically accommodate a wide range of fairness constraints, how it wraps around/uses group information, etc.
> >
> > (That being said, I expect that the authors will keep iterating on these improvements to further streamline the presentation of the framework and its merits. Here is one important such example, which relates to the authors' points in the rebuttal to my review regarding the novelty of the methodological contributions. I believe the definition of the notion of a "closeness criterion" should be discussed in a significantly more focused way *before* the algorithm, with a dedicated short paragraph delineating i.a. how it is responsible for controlling the miscoverage gap pairwise over all groups (and previewing that it will later be further investigated for e.g. predictive parity). Note that e.g. the second and third aspects of novelty discussed in the rebuttal are related to this notion. Currently, the updated manuscript discusses this in roundabout ways such as by referring to it as the "inverse quantile" technique, which can sound less interpretable to a fairness audience.)
> >
> > On the experimental side, the empirics are now more comprehensive. Additional baseline vanilla conformal approaches were added in as requested, alongside a comparison to a group-conditional conformal method that was amongst the previously omitted references, which are now also included. Correspondingly, the additional related work section has also been rephrased and expanded to reflect the questions in my and the other reviews.
> >
> > On the whole, I believe that in its updated state, the paper is now in the shape to be useful to the union/intersection of the conformal prediction and ML fairness audiences; thus my updated score is now on the positive side.

---

> ### Comment · Area_Chair_tojJ · 2024-11-24
>
> Dear Reviewer ST3N,
>
> The author discussion phase will be ending soon. The authors have provided detailed responses. Could you please reply to the authors with whether they have addressed your concerns and whether you will keep or modify your assessment on this submission?
>
> Thanks,
>
> Area Chair

---

> ### Author Response · Authors · 2024-11-28
> **Thank You**
>
> We thank the reviewer for engaging in a discussion about our work, for their constructive comments, and for updating their scores. We will iterate and improve the presentation while addressing specific suggestions (e.g., the closeness criterion discussion). We also thank the reviewer for acknowledging our work's usefulness for conference attendees interested in the union/intersection of conformal prediction and ML fairness. We are happy to answer any further questions the reviewer may have.

---

### Author Response · Authors · 2024-11-24
**Overview Response to Reviewers and Area Chairs**

We thank all the reviewers for taking the time to review our paper. We clarify the contributions and summarize the improvements to the presentation and the new results we have added.

### Contributions

Our work is focused on formalizing the notion of Conformal Fairness and presenting an algorithm and associated framework for practitioners to control for gaps in coverage between groups. Our framework differentiates itself from the existing literature in a few different aspects:

1. **_Group Information at Inference:_** Existing works in group conditional prediction are limited by the need for group information at inference time, which can be purposefully withheld. If group information is absent, the CF framework still provides group-conditional (mis)coverages that can be balanced for conformal fairness.

2. **_Balancing miscoverage for all groups and all labels:_** In our framework, fairness is formulated as the pairwise miscoverage disparity between groups. For certain notions of fairness (i.e., Equal Opportunity) we extend the fairness formulation to condition on the label being positive when measuring disparity.
   \
    \
   To balance the miscoverages between groups, we use the inverse quantile to determine the miscoverage for a given threshold. The inverse quantile operation is unique in this context and is not similarly utilized in the existing literature.

3. **_Framework Flexibility:_** Our work differs in the flexibility it provides a practitioner to explore and evaluate fairness on their own conformal prediction methods for common and user-defined fairness metrics. To demonstrate this flexibility, we consider a proxy-variant of Predictive Parity.

### Presentation Updates

Based on the feedback from all the reviewers (in particular Reviewer ST3N's), we have substantially tightened the writing of Section 3 to be more precise. Specific changes include:

- **_(Conformal) Fairness Metrics:_** We elaborate on how the standard fairness definitions are adapted to the conformal formulations. We will include descriptions of each fairness metric in Appendix A in a camera-ready version.
- **_Nomenclature:_** We introduce terms like “interval_width”, “filter_function”, and “label_miscoverage” earlier in the section, before presenting the algorithm, with more mathematical formalism (e.g. “filter_function” is presented as $F_M$) to reduce ambiguity and clunkiness. We will also include a symbols table in the Appendix.\
  \
  To further reduce possible ambiguity, we changed the phrase “closeness threshold” to “closeness criterion” to avoid overloading terminology, since threshold ($\lambda$) is also used when constructing the prediction sets.

Other writing changes we made in response to Reviewers ST3N, zT1T, VVws, and 3NSU are:

- Expanding the motivation/intuition of the theory with concrete examples.
- Clarifying Theorem 3.4 to more precisely state what is being controlled, when considering Predictive Parity.
- In Section 4.3, we include references and discussions for the papers that the reviewers have shared.

### New Results

Our paper is focused on making a fair version of conformal prediction, thus we compare it to standard CP methods. We demonstrate the framework’s effectiveness in improving those methods in terms of fairness. These results are presented in the main body and supplemental material. We also updated all of the figures and tables to include RAPS (as per ST3N).

We further note that existing works, shared by Reviewer ST3N, study group-conditional coverage guarantees they omit class conditionality which is a part of many fairness metrics. We include additional results adapting BatchGCP [1] as a threshold function for Demographic Parity (Figures E12 and E13) and Disparate Impact (Table E6) on the ACSIncome and ACSEducation datasets in Appendix E.6. The results demonstrate that the CF Framework outperforms methods in group-conditional CP in terms of fairness disparity, with a small cost to efficiency. For our other datasets (i.e. Pokec and Credit), the BatchGCP codebase requires additional changes to be accommodated.

For existing works that are at the intersection of fairness and conformal prediction, we note that some are an instantiation of our work [2], are too narrow of an application [3], or are designed for a different task or a single fairness metric [4] to act as a suitable baseline.

We have updated the main paper and parts of the supplementary to reflect all of the reviewer's comments, additional results, and the presentation as discussed above.
\
\
\
Again, we thank the reviewers for their detailed reviews and are happy to answer any further questions that may arise.

[1] C. Jung et al.: Batch Multivalid Conformal Prediction [ICLR 2023]\
[2] Wang et al.: Equal opportunity of coverage in fair regression. [NIPS 2023]\
[3] Lu et al.: Fair conformal predictors for applications in medical imaging. [AAAI 2022]\
[4] Liu et al.: Conformalized fairness via quantile regression. [NIPS 2022]

---

### Meta-Review · Area_Chair_tojJ · 2024-12-11

**Metareview:**

The paper introduces the concept of Conformal Fairness and proposes a novel algorithm to control coverage gaps between groups in predictive models. This paper makes an important contribution to the fairness issue of conformal learning. More specifically, compared to the existing methods, the proposed algorithm  does not require group information at inference time, which is big advantage. Also, the method is very flexible and can be used with different user-specified fairness metric. I recommend acceptance after reading the paper and the reviewers' feedback.

**Additional Comments On Reviewer Discussion:**

Reviewer ST3N expressed concerns on the writing and the theoretical novelty of this paper. The authors updated the paper to improve its exposition and explained the theoretical novelty. Then Reviewer ST3N raised the score from 5 to 6.

Reviewer zT1T had questions on the lack of the fairness-efficiency trade-off and the exchangeability assumption, but the authors have well clarified the questions.

Reviewer VVws's review is very short and has a lower confidence score, so I have to give a lower weight on his/her review.

Reviewer 3NSU thought there was a missing discussion about Theorem 3.4 and the choice of parameter $\alpha$. The authors have well addressed these questions as acknowledged by the reviewer.

---

### Decision · Program_Chairs · 2025-01-22

Accept (Poster)